# OwlEye: Zero-Shot Learner for Cross-Domain Graph Data Anomaly Detection

**Lecheng Zheng[1], Dongqi Fu[2], Zihao Li[1], Jingrui He[1]**
University of Illinois Urbana-Champaign[1], Meta AI[2]
{lecheng4, zihaoli5, jingrui}@illinois.edu, dongqifu@meta.com

## ABSTRACT

Graph data is informative to represent complex relationships such as transactions between accounts, communications between devices, and dependencies among machines or processes. Correspondingly, graph anomaly detection (GAD) plays a critical role in identifying anomalies across various domains, including finance, cybersecurity, manufacturing, etc. Facing the large-volume and multi-domain graph data, nascent efforts attempt to develop foundational generalist models capable of detecting anomalies in unseen graphs without retraining. To the best of our knowledge, the different feature semantics and dimensions of cross-domain graph data heavily hinder the development of the graph foundation model, leaving further in-depth continual learning and inference capabilities a quite open problem. Hence, we propose **OwlEye**, a novel zero-shot GAD framework that learns transferable patterns of normal behavior from multiple graphs, with a three-fold contribution. First, OwlEye proposes a **cross-domain feature alignment** module to harmonize feature distributions , which preserves domain-specific semantics during alignment. Second, with aligned features, to enable continuous learning capabilities, OwlEye designs the **multi-domain multi-pattern dictionary learning** to encode shared structural and attribute-based patterns. Third, for achieving the in-context learning ability, OwlEye develops a **truncated attention-based reconstruction** module to robustly detect anomalies without requiring labeled data for unseen graph-structured data. Extensive experiments on real-world datasets demonstrate that OwlEye achieves superior performance and generalizability compared to state-of-the-art baselines, establishing a strong foundation for scalable and label-efficient anomaly detection. The code is available at https://github.com/zhenglecheng/ICLR-2026-OWLEYE.

## 1 INTRODUCTION

Graph anomaly detection (GAD) has been extensively studied over the past decades due to its wide-ranging applications that naturally involve graph-structured data, such as transaction networks in financial fraud detection (Slipenchuk & Epishkina, 2019; Ramakrishnan et al., 2019), communication and access networks in cybersecurity intrusion detection (Brdiczka et al., 2012; Duan et al., 2023), and user-user interaction graphs in fake news detection on social networks (Shu et al., 2017; 2019; Fu et al., 2022a). Driven by the growing demand for accurate and scalable anomaly detection, recent research increasingly leverages graph neural networks (GNNs) to model node-level irregularities in complex graph-structured data. Broadly, existing approaches to GAD can be categorized into two research directions. The first adopts a "*one model for one dataset*" paradigm (Qiao & Pang, 2023; Zheng et al., 2025a; Liu et al., 2022; Huang et al., 2022; Qiao et al., 2025), where one single model is trained for each graph individually to detect anomalies within that specific context. While this strategy can be effective, it is often computationally expensive and suffers from limited generalizability to unseen graphs. In contrast, a new research direction aims to build "*one for all*" generalist frameworks (Niu et al., 2024; Liu et al., 2024) that are trained on multiple graphs and capable of detecting anomalies in entirely new, unseen graphs without retraining. These models offer advantages in terms of scalability and cross-domain adaptability. For example, ARC (Liu et al., 2024) introduces a generalist framework based on in-context learning, which encodes high-order affinity and heterophily into anomaly-aware embeddings transferable across datasets. UNPrompt (Niu et al.,

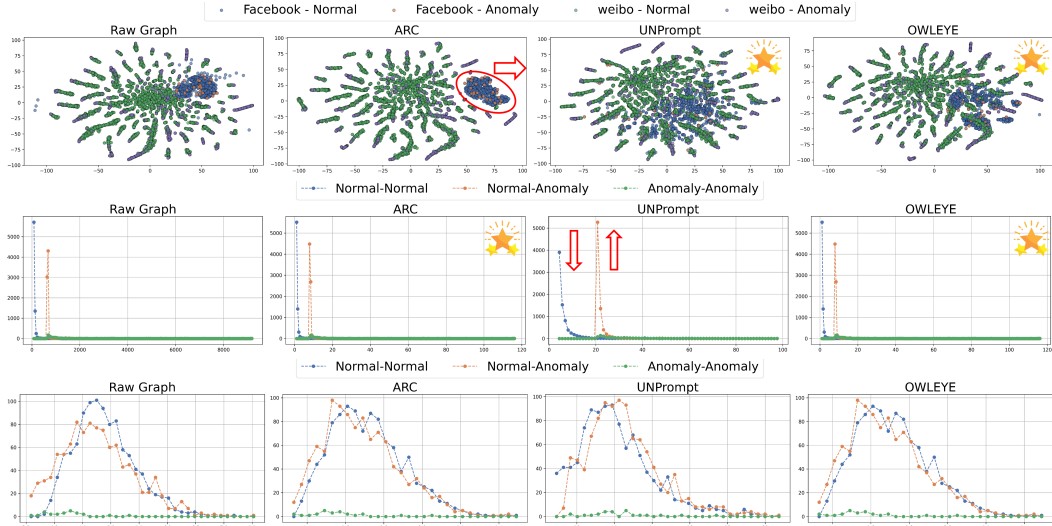

Figure 1: **Performance Visualization of SOTA GAD methods**, ☀ denotes best. *Top row:* TSNE embeddings of Facebook and Weibo graph data for (a) original features, (b) ARC, (c) UNPrompt, and (d) OWLEYE (ours). ARC pushes the two graphs apart rather than aligning them. *Middle and bottom rows:* pairwise Euclidean distances for Normal-Normal, Normal-Anomaly, and Anomaly-Anomaly node pairs on Weibo and Facebook dataset, respectively. In the original graph (middle row, (a)), Normal-Normal pairs are denser than Normal-Anomaly pairs on Weibo dataset—an important pattern reversed by UNPrompt (middle row, (c)). The existing data preprocessing methods fail to either align the graphs into the share space or preserve important patterns after normalization.

2024) proposes generalized neighborhood prompts that leverage latent node attribute predictability as an anomaly score, enabling effective anomaly detection in previously unseen graphs.

However, several critical limitations still hinder the full potential of these generalist models. **First**, graphs from different domains often have inherently different feature spaces and semantic interpretations. Existing methods (Niu et al., 2024; Liu et al., 2024; Qiao et al., 2025) typically use dimensionality reduction techniques such as principal component analysis (PCA) (Maćkiewicz & Ratajczak, 1993) or singular value decomposition (SVD) (Hoecker & Kartvelishvili, 1996), along with basic normalization strategies, to enforce a shared input space with the same dimensionality. However, these heuristics frequently fail to align heterogeneous feature distributions and preserve the important structural patterns effectively. In Figure 1, we visualize TSNE embeddings of Facebook and Weibo graphs before and after preprocessing with three methods (top row), along with pairwise Euclidean distances among three types of node pairs: Normal-Normal, Normal-Anomaly, and Anomaly-Anomaly for both datasets (middle and bottom rows). Notably, the TSNE plots show that ARC (Liu et al., 2024) tends to separate the two graphs rather than align them, while UN-Prompt (Niu et al., 2024) disrupts critical structural patterns, for instance, it reverses the density relationship between Normal-Normal and Normal-Anomaly pairs observed in the original graph (compare subfigures in the middle row columns (a) and (c)). **Second**, current approaches lack mechanisms for continual capabilities as they do not support seamless integration of new graphs and the incremental update of normal and abnormal patterns without retraining from scratch. **Third**, many existing models assume the availability of a few labeled nodes in the target graph to facilitate few-shot learning. In practice, however, labeling anomalies can be costly and requires domain expertise, making this assumption unrealistic. This raises an important question: *how can we enable zero-shot anomaly detection without relying on any labeled data from the test graph?*

To address these limitations, we propose **OWLEYE**, a novel generalist for zero-shot graph anomaly detection across multiple domains. In brief, the core idea of OWLEYE is to learn and store representative patterns of normal behavior from multiple source graphs in a structured dictionary that acts as a knowledge base. When applied to an unseen graph, OWLEYE can effectively detect anomalies by leveraging the representative patterns stored in the dictionary. Our approach is built on three key components. **First**, we introduce a cross-domain feature alignment module, which normalizes

and aligns node features across graphs using pairwise distance statistics, ensuring that graphs from different domains can be embedded into a shared input feature space. **Second**, we develop a multi-domain pattern module that extracts both attribute-level and structure-level patterns from training graphs and stores them in a pattern dictionary. This dictionary enables the model to generalize to unseen graphs together with the cross-domain feature alignment module. **Third**, we design a truncated attention-based cross-domain reconstruction module that samples a subset of nodes and reconstructs them using the stored patterns, effectively identifying anomalies while minimizing the influence of abnormal nodes during the reconstruction process.

## 2 OwlEye: Zero-Shot Cross-Domain Graph Anomaly Detector

In this section, we present OwlEye and illustrate cross-domain feature alignment, multi-domain pattern learning, and truncated attention-based reconstruction. Throughout this paper, we use regular letters to denote scalars (*e.g.*, $\alpha$), boldface lowercase letters to denote vectors (*e.g.*, $\boldsymbol{x}$), and boldface uppercase letters to denote matrices (*e.g.*, $\boldsymbol{A}$). Let $\mathcal{G} = (\mathcal{V}, \mathcal{E}, \boldsymbol{X})$ be an undirected graph, where $\mathcal{V}, \mathcal{E}, \boldsymbol{X}$ are the set of nodes, set of edges and the node attribute matrix, respectively. Let $\mathcal{T}_{train} = \{\mathcal{D}_{train}^1, \mathcal{D}_{train}^2, ..., \mathcal{D}_{train}^m\}$ be a set of training datasets with $m$ graphs and each $\mathcal{D}_{train}^i = (\mathcal{G}_{train}^i, \boldsymbol{y}_{train}^i)$ is a labeled dataset, where $\boldsymbol{y}_{train}^i$ is a label vector denoting the abnormality of each node in the graph $\mathcal{G}_{train}^i$. Our objective is to train a GAD model on $\mathcal{T}_{train}$ to identify anomalous nodes in the graph $\mathcal{T}_{test}^j$ from the test datasets $\mathcal{T}_{test} = \{\mathcal{D}_{test}^1, \mathcal{D}_{test}^2, ..., \mathcal{D}_{test}^{m'}\}$, where $m'$ represents the number of graphs in the test datasets.

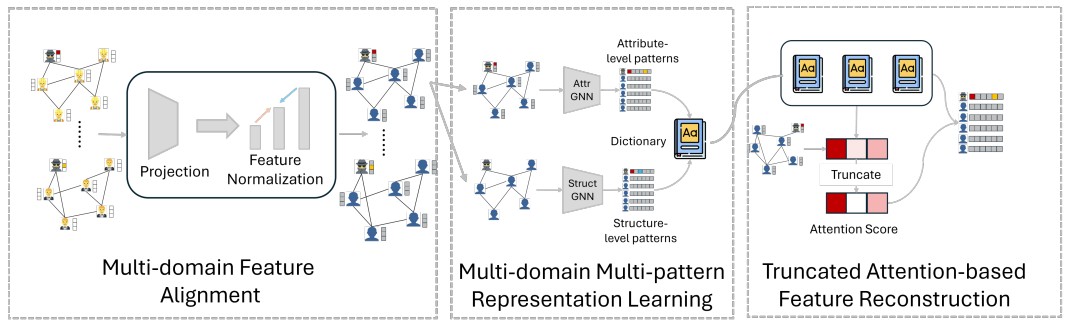

Figure 2: Overview of OwlEye.

### 2.1 Preserving Domain-specific Semantics in Cross-domain Feature Alignment

Graph data from different domains often exhibit heterogeneous features that vary in both dimensionality and semantic meaning. For example, nodes in a citation network may be described by textual content and paper metadata, whereas nodes in a social network might be characterized by user profile attributes. This heterogeneity poses a significant challenge for generalist GAD models, which require a consistent input representation across domains. Therefore, how can we effectively unify and align heterogeneous features from diverse graph domains without compromising their semantic integrity? To address this issue, we first propose to project these features into a common feature space to achieve the consistent feature dimension and then employ the cross-domain feature alignment to align the features from different graphs into a shared input space without compromising their semantic integrity.

**Feature Projection.** To achieve the consistent attribute dimension across different graphs, we employ the principal component analysis (PCA) technique on the raw features of each graph $\mathcal{G}^i \in \mathcal{T}_{train} \cup \mathcal{T}_{test}$. Specifically, given the attribute matrix $\boldsymbol{X}^i \in \mathbb{R}^{n^i \times d^i}$ from the graph $\mathcal{G}^i$ with $n^i$ nodes, we aim to transform it to $\tilde{\boldsymbol{X}}^i \in \mathbb{R}^{n^i \times d}$ with the common dimensionality of $d$ by:

$$\tilde{\boldsymbol{X}}^i = \text{Proj}(\boldsymbol{X}^i) \tag{1}$$

where $\text{Proj}(\cdot)$ is PCA.

**Cross-domain Feature Normalization.** Although PCA enables consistent dimensionality across graphs, the semantic meaning of each projected feature across different datasets remains distinct. As

shown in Figure 1, the TSNE plots show that ARC (Liu et al., 2024) tends to separate the two graphs rather than align them, while UNPrompt (Niu et al., 2024) distorts the input space by reversing the density relationship between Normal-Normal and Normal-Anomaly pairs observed in the original graph (compare subfigures (a) and (c) in the second row). To align attributes from different graphs into a shared input space, we propose a cross-domain feature Normalization to align the semantics and unify the distributions across graphs. Specifically, we first compute the average normalization of node features in the $i$-th graph $\mathcal{G}^i$ by $N^i = \frac{1}{n^i} \sum_{j \in \mathcal{V}_i} \sqrt{\sum_{k=1}^{d_i} (\tilde{\boldsymbol{x}}_{jk}^i)^2}$.

In the second row of Figure 1, we observe that a large distance gap between a Normal-Normal pair and a Normal-Anomaly pair on the Weibo dataset can be a crucial pattern for a GAD model to find a proper decision boundary to separate normal and anomalous nodes. This observation inspires us to leverage the pairwise node distance for Normal-Normal pairs, Normal-Anomaly pairs and Anomaly-Anomaly pairs. However, the lack of label information for a graph from the test set prevents us from fully benefiting from this observation. To relax the constraint of the unavailable label information, we propose to measure the pairwise distance as an important indicator for preserving the structural pattern in the latent space regardless of its label information. Specifically, we measure the pairwise node distance for the original graph and the normalized graph as follows:

$$\text{dist}^i = \frac{1}{(n^i)^2} \sum_{v_j, v_k \in \mathcal{V}_i} \sqrt{(\tilde{\boldsymbol{x}}_{v_j}^i - \tilde{\boldsymbol{x}}_{v_k}^i)^2}, \ \text{dist}_N^i = \frac{1}{(n^i)^2} \sum_{v_j, v_k \in \mathcal{V}_i} \sqrt{(\frac{\tilde{\boldsymbol{x}}_{v_j}^i}{N^i} - \frac{\tilde{\boldsymbol{x}}_{v_k}^i}{N^i})^2} \tag{2}$$

where $\text{dist}^i$ and $\text{dist}_N^i$ measure the distance between the pairwise node distance for the original graph $\mathcal{G}^i$ and the normalized graph, respectively. To utilize the pairwise distance for the graphs from the training set, we take the median over all available graphs.

$$\text{dist}^{\text{med}} = \text{median}([\text{dist}^1, \text{dist}^2, ..., \text{dist}^m]), \text{dist}_N^{\text{med}} = \text{median}([\text{dist}_N^1, \text{dist}_N^2, ..., \text{dist}_N^m]) \tag{3}$$

The reason why we choose not to use the average operation is to avoid a situation where a graph with too large average pairwise distance dominates the datasets. Finally, we normalize the node attributes in the $i$-th graph as follows:

$$\tilde{X}^i \leftarrow \frac{\tilde{X}^i}{N^i} \cdot \max(f, \tau) \tag{4}$$

where $f = \sqrt{\frac{\text{dist}^{\text{med}} \cdot \text{dist}_N^i}{\text{dist}^i \cdot \text{dist}_N^{\text{med}}}}$ is a scaling factor to control the magnitude of the pairwise distance cross different graphs, and $\tau$ is the constant positive temperature, which is set to 1 in the experiments.

## 2.2 Multi-domain Multi-pattern Dictionary Learning

One major limitation of existing generalist GAD models is their inability to support the seamless integration of knowledge extracted from new graphs, as well as the incremental updating of normal and abnormal patterns without retraining from scratch. To this end, can we design a generalist GAD framework that enables continual learning by efficiently accumulating and updating knowledge across diverse graph domains? Hence, we propose to learn and extract both attribute-level and structure-level patterns and store them in a *dynamic dictionary*.

**Attribute-level Representation Learning.** Following the existing generalist GAD methods (Niu et al., 2024; Liu et al., 2024; Qiao et al., 2025), we aim to learn and extract the generalized attribute-level $\boldsymbol{H}_{\text{attr}}^{i,l+1}$ representations across graphs:

$$\boldsymbol{H}_{\text{attr}}^{i,l+1} = \sigma(\boldsymbol{A}^i \boldsymbol{H}_{\text{attr}}^{i,l} \boldsymbol{W}_{\text{attr}}^l) \tag{5}$$

where $\boldsymbol{A}^i \in \mathbb{R}^{n^i \times n^i}$ is the adjacency matrix of graph, $\boldsymbol{W}_{\text{attr}}^l \in \mathbb{R}^{d \times d}$ is the learnable weight matrix of the $l$-th layer graph neural network (Kipf & Welling, 2017) to learn the attribute representations, $\boldsymbol{H}_{\text{attr}}^{i,l} \in \mathbb{R}^{n^i \times d}$ denotes the attribute-level embedding of the graph $\mathcal{G}^i$ and $\boldsymbol{H}_a^{i,0} = \tilde{X}^i$. To fully capture the high-order neighborhood information, we concatenate the multi-hop information with the residual network (He et al., 2016):

$$\boldsymbol{H}^i = [\boldsymbol{H}_{\text{attr}}^{i,2} - \boldsymbol{H}_{\text{attr}}^{i,1}, \ldots, \boldsymbol{H}_{\text{attr}}^{i,l+1} - \boldsymbol{H}_{\text{attr}}^{i,1}] \tag{6}$$

where $\boldsymbol{H}^i \in \mathbb{R}^{n^i \times ld}$.

**Structure-level Representation Learning.** The attribute-level representation $\boldsymbol{H}^i$ captures node-specific information (e.g., a user's interests in a social network), while it fails to capture relationships and interactions (e.g., who connects to whom and how densely). For example, in social networks, two users may have similar attributes but vastly different roles based on their connectivity (e.g., normal users vs. anomalous users). Thus, a natural question arises: can we learn the structural representations without the intervention of the node attributes? Inspired by this, we propose to first replace the raw node attributes with a $d$-dimension all-one vector and learn structural representations $\boldsymbol{R}_{\text{struc}}^{i,l+1}$ as follows:

$$\boldsymbol{R}_{\text{struc}}^{i,l+1} = \sigma(\boldsymbol{A}^i \boldsymbol{R}_{\text{struc}}^{i,l} \boldsymbol{W}_{\text{struc}}^l) \tag{7}$$

where $\boldsymbol{W}_s^l \in \mathbb{R}^{d \times d}$ is the learnable weight matrix of the $l$-th layer graph neural network to learn the structural representations, $\boldsymbol{R}_{\text{struc}}^{i,l} \in \mathbb{R}^{n^i \times d}$ denotes the structural embedding of the graph $\mathcal{G}^i$ and $\boldsymbol{R}_{\text{struc}}^{i,0} = \mathbf{1} \in \mathbb{R}^{n^i \times d}$ is the all-one matrix. Similarly, we concatenate the multi-hop information with the residual network to fully capture the high-order neighborhood information $\boldsymbol{R}^i$:

$$\boldsymbol{R}^i = [\boldsymbol{R}_{\text{struc}}^{i,2} - \boldsymbol{R}_{\text{struc}}^{i,1}, \ldots, \boldsymbol{R}_{\text{struc}}^{i,l} - \boldsymbol{R}_{\text{struc}}^{i,1}] \tag{8}$$

**Cross-domain Pattern Extraction.** After learning both attribute-level representation $\boldsymbol{H}$ and structure-level representation $\boldsymbol{R}$ for even available graph, we randomly extract $n_{sup}$ patterns in total from each graph $\mathcal{G}^j$ and store them in two dictionaries by:

$$\text{Dict}_H^j = \boldsymbol{H}^j[\text{idx}^j], \ \text{Dict}_R^j = \boldsymbol{R}^j[\text{idx}^j] \tag{9}$$

where $\text{idx}^j$ is a set of the node index randomly sampled from the graph $\mathcal{G}^j$. Due to the fact that the patterns extracted from different graphs contribute differently for detecting anomalies in these graphs, we propose to measure the similarity of nodes between graph $\mathcal{G}^i$ and the patterns $\text{Dict}_R^j$ extracted from graph $\mathcal{G}^j$ only based on the structure-level representation as follows:

$$\text{sim}(\mathcal{G}^i, \text{Dict}_R^j) = \max(\text{softmax}(\boldsymbol{R}^i \boldsymbol{W}_1 (\boldsymbol{R}^j[\text{idx}])^T)) \tag{10}$$

where $\text{sim}(\mathcal{G}^i, \text{Dict}_R^j) \in \mathbb{R}^{n^i}$ measures the maximal node similarity between graph $\mathcal{G}^i$ and the patterns stored in $\text{Dict}_R^j$, and $\boldsymbol{W}_1 \in \mathbb{R}^{ld \times ld}$ is a weight matrix. Here, we expand $\text{sim}(\mathcal{G}^i, \text{Dict}_R^j) \in \mathbb{R}^{n^i}$ to be $\mathbb{R}^{n^i \times ld}$. The reason why we only use structure-level representation for similarity measurement is that leveraging attribute-level representation may fail to distinguish the camouflaged anomalous nodes with the attributes similar to its normal neighbors. Notice that the advantages of extracting and storing the patterns in a dictionary include that these representative patterns could be leveraged for anomaly detection in the unseen graphs and that the dictionary could be easily updated by adding more patterns, thus enabling the continual evolving capabilities of OWLEYE. Similarly, $\text{sim}(\mathcal{G}^i, \text{Dict}_H^j)$ can be computed via Eq.10 by replacing the structure-level representation $\boldsymbol{R}$ with attribute-level representation $\boldsymbol{H}$. Three case studies in Section 3.4 verify that OWLEYE has excellent continual evolving capabilities and achieves better performance on test graphs on average by directly adding more patterns extracted from other new graphs to the dictionary without retraining or finetuning the model.

## 2.3 TRUNCATED ATTENTION-BASED FEATURE RECONSTRUCTION FOR IN-CONTEXT LEARNING AND INFERENCE

Many existing models (Niu et al., 2024; Liu et al., 2024; Qiao et al., 2025) usually assume the availability of a few labeled nodes in the test graph to facilitate few-shot learning. However, labeling anomalies can be costly and requires domain expertise in practice, making this assumption unrealistic. When labels are hardly available, how can we enable zero-shot anomaly detection without relying on any labeled data from the test graph? A naive solution is to randomly sample pseudo-support nodes from a test graph as normal nodes due to the fact that the vast majority of the nodes are normal.

However, it inevitably leads to an issue that abnormal nodes might be misleadingly labeled as the pseudo-support nodes, thus resulting in performance degradation. To address this issue, we propose a truncated attention-based reconstruction method to filter out the potential abnormal nodes and only

select the most representative nodes to reconstruct both attribute-level representation and structure-level representation. Specifically, we propose to measure the truncated attention score for the query node from the graph $\mathcal{G}^i$ and the normal attribute-level patterns $\boldsymbol{H}^j$ from $\text{Dict}_H^j, j \in \{1, ..., m\}$ by:

$$
\alpha(\mathcal{G}^i, \text{Dict}_H^j) = \sqrt{\frac{(\boldsymbol{W}^Q \boldsymbol{H}^i)(\boldsymbol{W}^K (\boldsymbol{H}^j)^T)}{\sqrt{ld}}}
$$

$$
\alpha(\mathcal{G}^i, \text{Dict}_H^j)[\text{idx}_\alpha] = -\infty, \text{ where } \text{idx}_\alpha = \text{Top}(-\alpha(\mathcal{G}^i, \text{Dict}_H^j), k)
$$

$$
\alpha_H^{ij} = \text{softmax}(\alpha(\mathcal{G}^i, \text{Dict}_H^j)/\tau_a)
$$

(11)

where $\boldsymbol{W}^Q \in \mathbb{R}^{ld \times ld}$, $\boldsymbol{W}^K \in \mathbb{R}^{ld \times ld}$ are two weight matrices shared across all $i$ and $j$ pairs, $\text{Top}(\cdot, k)$ selects the top $k$ node indices to be truncated, and $\tau_a$ is the temperature magnifying the significance of the selected patterns. Notice that $\alpha(\mathcal{G}^i, \text{Dict}_H^j)[\text{idx}_\alpha] = -\infty$ truncates the less representative nodes for the attribute-level representation reconstruction after softmax operation. Similarly, we can compute the attention $\alpha_R^{ij}$ for structure-level patterns $\boldsymbol{R}^j$ from $\text{Dict}_R^j$ following Equation 11.

Then, we reconstruct both the attribute-level and structure-level representation in the graph $\mathcal{G}^i$ with the normal patterns from $m$ training graphs by:

$$
\hat{\boldsymbol{H}}^i = \frac{1}{m} \sum_{j=1}^m \text{sim}(\mathcal{G}^i, \text{Dict}_H^j) \odot (\alpha_H^{ij} \text{Dict}_H^j)
$$

$$
\hat{\boldsymbol{R}}^i = \frac{1}{m} \sum_{j=1}^m \text{sim}(\mathcal{G}^i, \text{Dict}_R^j) \odot (\alpha_R^{ij} \text{Dict}_R^j)
$$

(12)

where $\odot$ denotes the Hadamard product. The intuition is that we aim to use the normal patterns extracted from each training graph to reconstruct the node embedding in the graph $\mathcal{G}^i$.

**Training.** To optimize OWLEYE on training sets $\mathcal{T}_{train}$, we aim to minimize the following objective function:

$$
\mathcal{L}_{\text{recon}} = \sum_{i=1}^m \sum_{v_k \in \mathcal{A}^i} \frac{\boldsymbol{H}_{v_k}^i (\hat{\boldsymbol{H}}_{v_k}^i)^T}{|\boldsymbol{H}_{v_k}^i||\hat{\boldsymbol{H}}_{v_k}^i|} - \sum_{v_j \in \mathcal{N}^i} \frac{\boldsymbol{H}_{v_j}^i (\hat{\boldsymbol{H}}_{v_j}^i)^T}{|\boldsymbol{H}_{v_j}^i||\hat{\boldsymbol{H}}_{v_j}^i|}
$$

$$
\mathcal{L}_{\text{triplet}} = \sum_{i=1}^m \sum_{v_j \in \mathcal{A}^i, v_k \in \mathcal{N}^i} [\max(||\hat{\boldsymbol{H}}_{v_j}^i - \boldsymbol{H}_{v_j}^i||^2 - ||\hat{\boldsymbol{H}}_{v_j}^i - \hat{\boldsymbol{H}}_{v_k}^i||^2 + \lambda, 0)
$$

(13)

$$
+ \beta \max(||\hat{\boldsymbol{R}}_{v_j}^i - \boldsymbol{R}_{v_j}^i||^2 - ||\hat{\boldsymbol{R}}_{v_k}^i - \hat{\boldsymbol{R}}_{v_k}^i||^2 + \lambda, 0)]
$$

$$
\mathcal{L} = \mathcal{L}_{\text{triplet}} + \mathcal{L}_{\text{recon}}
$$

Where $\mathcal{N}^i = \{v_j | y_j = 0\}$ denotes the set of normal nodes, $\mathcal{A}^i = \{v_k | y_k = 1\}$ is the set of anomalous nodes and $\lambda$ is the margin of the triplet loss. In $\mathcal{L}_{\text{recon}}$, we minimize the instance-wise difference (maximize the similarity) between the attribute-level representation $\boldsymbol{H}_{v_j}^i$ and the reconstructed attribute-level representation $\hat{\boldsymbol{H}}_{v_j}^i$ for $v_j \in \mathcal{N}$ but maximize the difference (minimize the similarity) between the attribute-level representation $\boldsymbol{H}_{v_j}^i$ and the reconstructed attribute-level representation $\hat{\boldsymbol{H}}_{v_k}^i$ for $v_j \in \mathcal{N}$ and $v_k \in \mathcal{A}$. Compared with $\mathcal{L}_{\text{recon}}$, minimizing $\mathcal{L}_{\text{triplet}}$ allows more pairwise contrasting (e.g., $||\hat{\boldsymbol{H}}_{v_j}^i - \hat{\boldsymbol{H}}_{v_k}^i||^2$) for robust representation learning.

**Inference.** At the inference stage, we first extract $n_{sup}$ normal patterns stored in the dictionaries $\text{Dict}_H$ and $\text{Dict}_R$ for each graph from the training set $\mathcal{T}_{train}$. Given a graph $\mathcal{G}^i$ from the test set $\mathcal{T}_{test}$, we extract and store $n_{sup}$ normal patterns from $\mathcal{G}^i$ in the dictionaries $\text{Dict}_H$ and $\text{Dict}_R$ for representation reconstruction by Equation (12). The anomaly score of node $v_j$ is computed as follows:

$$
\mathcal{S}_{v_j} = ||\hat{\boldsymbol{H}}_{v_j}^i - \boldsymbol{H}_{v_j}^i||^2 + \beta ||\hat{\boldsymbol{R}}_{v_j}^i - \boldsymbol{R}_{v_j}^i||^2
$$

(14)

## 3 EXPERIMENTS

### 3.1 EXPERIMENTAL SETUP

**Datasets.** In the experiments, we train OWLEYE and the baseline methods on a group of graph datasets and test on another group of datasets. Following (Liu et al., 2024), the training graphs span across a variety of domains, including social networks, citation networks, and e-commerce co-review networks, each of them with either injected anomalies or real anomalies. Therefore, we select one graph from each domain and randomly select one more dataset in the training set (e.g., CiteSeer). Specifically, the training datasets $\mathcal{T}_{train}$ consist of PubMed, CiteSeer, Questions, and YelpChi, while the testing datasets $\mathcal{T}_{test}$ consist of Cora, Flickr, ACM, BlogCatalog, Facebook, Weibo, Reddit, and Amazon. We also include the experimental results for different train-test split in Appendix B.2.

**Baselines.** We compare our proposed method with supervised methods, unsupervised methods and one-for-all methods. Supervised methods include two state-of-the-art GNNs specifically designed for GAD, i.e., BWGNN (Tang et al., 2022), and GHRN (Gao et al., 2023). Unsupervised methods include four representative approaches with distinct designs, including the generative method DOMINANT (Ding et al., 2019), the contrastive method SLGAD (Zheng et al., 2021b), two affinity-based methods (e.g., TAM (Qiao & Pang, 2023) and CARE (Zheng et al., 2025a)), one-for-all methods include ARC (Liu et al., 2024) and UNPrompt (Niu et al., 2024).

**Implementation Details.** Following (Liu et al., 2024; Qiao & Pang, 2023; Zheng et al., 2025a; Liu et al., 2022), we evaluate the performance of OWLEYE and baseline methods with respect to AUROC and AUPRC as evaluation metrics for GAD. We report the average AUROC/AUPRC with standard deviations across 5 trials. We train ARC on all the datasets in $\mathcal{T}_{train}$ jointly, and evaluate the model on each dataset in $\mathcal{T}_{test}$ in a zero shot setting. In the experiment, we set $\tau = 1$, $\tau_a = 0.001$, $n_{sup} = 2000$, $\lambda = 0.2$, and $\beta = 0.01$.

Table 1: Anomaly detection performance w.r.t AUPRC. We highlighted the results ranked first and second. "Average" indicates the average AUPRC over 8 datasets.

| Method | Cora | Flickr | ACM | BlogCatalog | Facebook | Weibo | Reddit | Amazon | Average |
|---|---|---|---|---|---|---|---|---|---|
| *Supervised (10-shot)* | | | | | | | | | |
| BWGNN | 9.57±2.40 | 12.39±2.68 | 13.37±6.03 | 12.97±3.15 | 5.81±1.17 | 9.55±2.12 | 3.21±2.32 | 12.40±1.86 | 9.80±2.91 |
| GHRN | 14.04±0.73 | 16.45±2.59 | 16.29±1.41 | 13.58±2.19 | 6.24±1.12 | 17.51±1.52 | 4.44±1.15 | 13.84±2.63 | 12.80±1.67 |
| *Unsupervised (zero-shot)* | | | | | | | | | |
| DOMINANT | 31.77±0.34 | 28.76±1.52 | 32.49±4.97 | 29.51±3.44 | 3.42±0.86 | 29.63±0.86 | 3.28±0.37 | 36.80±8.37 | 24.46±3.11 |
| SLGAD | 18.27±1.01 | 16.93±8.20 | 1.33±0.23 | 9.47±3.00 | 0.93±0.23 | 35.80±1.41 | 4.00±2.27 | 5.33±2.20 | 11.51±2.38 |
| TAM | 9.43±0.27 | 23.34±1.42 | 40.68±2.58 | 25.59±4.76 | 12.18±3.14 | 23.01±15.14 | 4.22±0.22 | 45.26±4.34 | 22.96±3.96 |
| CARE | 35.12±0.23 | 25.64±0.16 | 37.76±0.35 | 25.06±0.10 | 5.52±0.34 | 40.70±0.74 | 3.17±0.17 | 56.76±1.44 | 28.72±0.44 |
| ARC | 45.20±1.08 | 35.13±0.20 | 39.02±0.08 | 33.43±0.15 | 4.25±0.47 | 64.18±0.68 | 4.20±0.25 | 20.48±6.89 | 30.74±1.23 |
| UNPrompt | 9.84±2.90 | 25.21±1.84 | 11.18±1.67 | 18.24±13.05 | 4.32±0.55 | 20.58±5.62 | 3.77±0.32 | 9.41±2.69 | 12.82±3.58 |
| OWLEYE | 43.94±0.46 | 37.69±0.25 | 39.75±0.13 | 34.99±0.31 | 5.62±1.17 | 60.90±0.21 | 4.25±0.11 | 62.20±3.18 | 36.17±0.73 |

Table 2: Anomaly detection performance in 10-shot setting w.r.t AUPRC. We highlighted the results ranked first and second. "Average" indicates the average AUPRC over 8 datasets.

| Method | Cora | Flickr | ACM | BlogCatalog | Facebook | Weibo | Reddit | Amazon | Average |
|---|---|---|---|---|---|---|---|---|---|
| *Supervised (10-shot)* | | | | | | | | | |
| BWGNN | 9.57±2.40 | 12.39±2.68 | 13.37±6.03 | 12.97±3.15 | 5.81±1.17 | 9.55±2.12 | 3.21±2.32 | 12.40±1.86 | 9.80±2.91 |
| GHRN | 14.04±0.73 | 16.45±2.59 | 16.29±1.41 | 13.58±2.19 | 6.24±1.12 | 17.51±1.52 | 4.44±1.15 | 13.84±2.63 | 12.80±1.67 |
| *Unsupervised & Finetune (10-shot)* | | | | | | | | | |
| DOMINANT | 22.35±0.81 | 30.42±1.35 | 24.76±0.84 | 34.82±0.78 | 4.12±0.23 | 78.63±1.28 | 4.18±0.64 | 8.86±0.69 | 26.02±0.83 |
| SLGAD | 19.38±1.46 | 17.46±5.62 | 5.33±1.39 | 11.67±2.22 | 3.81±0.23 | 36.23±1.41 | 4.32±2.13 | 7.69±2.82 | 13.24±2.16 |
| TAM | 14.27±0.65 | 27.68±1.45 | 57.32±5.26 | 27.49±1.09 | 11.73±2.34 | 26.78±0.28 | 3.67±0.16 | 52.62±3.17 | 27.70±1.80 |
| CARE | 39.52±2.90 | 27.19±0.24 | 38.12±0.43 | 27.75±0.66 | 5.86±0.69 | 44.37±0.96 | 3.62±0.15 | 59.51±1.28 | 30.74±0.91 |
| ARC | 48.02±0.83 | 37.15±0.24 | 39.13±0.15 | 34.20±0.27 | 4.92±0.75 | 63.83±2.55 | 4.32±0.16 | 21.90±6.32 | 31.68±1.41 |
| UNPrompt | 11.40±0.55 | 22.65±0.30 | 14.80±0.70 | 18.01±12.88 | 4.04±1.07 | 22.23±4.83 | 3.85±0.21 | 11.08±0.46 | 13.51±2.62 |
| OWLEYE | 44.40±1.58 | 38.32±0.15 | 39.16±0.10 | 35.00±0.34 | 6.83±1.38 | 64.14±1.25 | 4.96±0.10 | 63.02±5.71 | 36.73±1.33 |

### 3.2 EXPERIMENTAL RESULTS

**Effectiveness Analysis.** In the experiment, we evaluate our method in the zero-shot and 10-shot setting on the graph from the test set $\mathcal{T}_{test}$. For the supervised learning methods (e.g., BWGNN and GHRN), we evaluate these two methods in the 10-shot setting, where we randomly sample 5 normal nodes and 5 anomalies as the support nodes. Table 1 and Table 2 show the performance of OWLEYE

and the baseline methods with respect to AUPRC in the zero-shot and 10-shot setting, respectively. The evaluation with respect to AUROC can be found in Tables 6 and 7 Appendix B.1. Based on the results, we have the following observations: (1). OWLEYE demonstrates strong anomaly detection capability in the generalist GAD scenario, without any finetuning. The average AUPRC is 5% higher than the best competitor (i.e., ARC). (2). Even if two supervised methods (e.g., BWGNN and GHRN) are provided with 10-shot label information, our proposed method OWLEYE can still outperform these two methods on 6 out of 8 datasets with respect to both AUPRC and AURPC. (3). If all methods are provided with 10-shot label information, OWLEYE achieves state-ofthe-art performance on 4 out of 8 in terms of AURPC and outperform the best competitor by more than 5%.

**Ablation Study.** We assess the effectiveness of three key components in OWLEYE (e.g., cross-domain feature normalization, structural pattern learning, and the truncated attention module) by comparing it with three variants: OWLEYE-N (without feature normalization), OWL-EYE-S (without structural patterns), and OWLEYE-T (with standard attention instead of truncated attention). All methods are trained on the same set $\mathcal{T}_{train}$ and evaluated on the test set $\mathcal{T}_{test} = \{$Cora, Flickr, ACM, BlogCatalog, Facebook, Weibo, Reddit, Amazon$\}$. Figure 3 (Left) shows the average AUPRC across these eight datasets. OWLEYE consistently outperforms all variants, highlighting the importance of each component in achieving robust cross-domain graph anomaly detection.

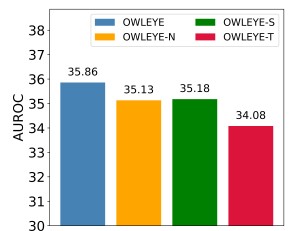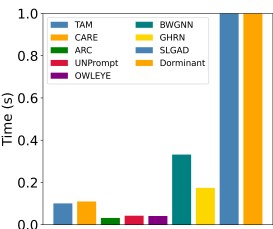

Figure 3: **Left:** Ablation Study. **Right:** Efficiency Analysis.

**Efficiency Analysis.** To evaluate the runtime efficiency of OWLEYE, we compare the finetuning time of different methods on the ACM dataset. As shown in Figure 3 (Right), OWLEYE achieves comparable finetuning time to other generalist GAD models, while significantly outperforming both unsupervised and supervised baselines (e.g., TAM, CARE, BWGNN, GHRN) in terms of efficiency.

### 3.3 VISUALIZATION OF CROSS-ATTENTION MAP ON CORA DATASETS

In this subsection, we provide a detailed visualization of the label matrices and cross-attention maps for the Cora dataset to better understand how our model distinguishes normal nodes from anomalous ones.

In Figure 4, subfigures (a) and (b) display the cross-attention scores across the graphs, where the y-axis corresponds to the graph indices (0–3 indicating the four training graphs and 4 representing the test graph) and the x-axis denotes the ten extracted patterns learned for each graph. The top row shows the attribute attention and structural attention for a normal node and the bottom row shows the attribute attention and structural attention for an anomalous one. Subfigure (a) shows the attention map when the model makes the correct prediction, while Subfigure (b) presents the attention map when the model makes the incorrect prediction. Subfigure (c) in both figures presents the ground-truth label matrices that specify whether each pattern is normal or abnormal. Across both datasets, lighter colors such as light green and light yellow consistently indicate high similarity to the patterns associated with normal nodes, as reflected in the label matrices in Subfigure (c).

By examining these visualizations, we observe a clear and consistent relationship between the attention intensity and the correctness of the model's predictions: when the model correctly identifies a normal node, its attention map is dominated by light colors, suggesting a strong similarity to normal patterns stored in the dictionary; when it correctly identifies an anomalous node, the attention map becomes noticeably darker, indicating low similarity to normal behavior. Importantly, this trend reverses for misclassified nodes: normal nodes that are wrongly predicted as anomalies exhibit darker color in attention map, while misclassified anomalous nodes show lighter colors, showing the high similarity to those of normal nodes. This systematic behavior demonstrates that the attention map offers an intuitive and faithful interpretation mechanism, as the color patterns directly reflect whether the node under consideration resembles the learned normal patterns, thereby revealing both the reasoning behind correct predictions and the failure modes behind incorrect ones.

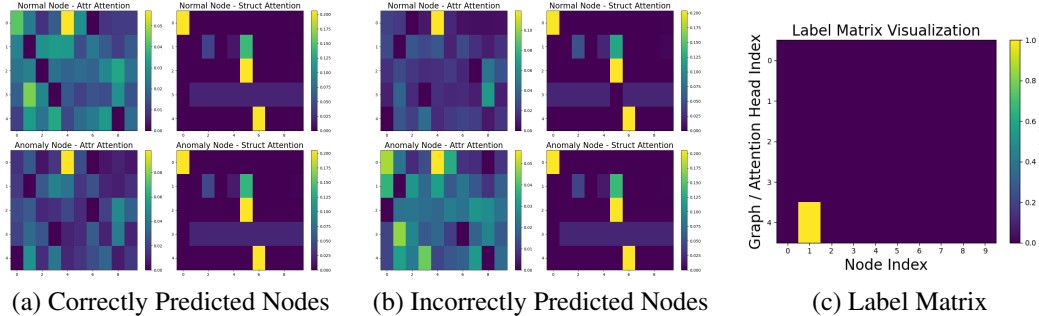

| (a) Correctly Predicted Nodes | (b) Incorrectly Predicted Nodes | (c) Label Matrix |

Figure 4: Visualization of cross-attention map on the Cora dataset. Subfigures (a) and (b) display the cross-attention scores across the graphs, where the y-axis corresponds to the graph indices (0–3 indicating the four training graphs and 4 representing the test graph) and the x-axis denotes the ten extracted patterns learned for each graph. The top row shows the attribute attention and structural attention for a normal node and the bottom row shows the attribute attention and structural attention for an anomalous one. Subfigure (c) in both figures presents the ground-truth label matrices that specify whether each node is normal or abnormal.

### 3.4 CASE STUDIES: ANALYSIS OF OWLEYE'S CONTINUAL LEARNING CAPABILITIES.

In this subsection, we present three case studies to evaluate how our proposed multi-domain pattern learning enhances OWLEYE's continual learning capability. Specifically, we consider the training set $\mathcal{T}_{train}$ = {PubMed, Cora, Questions, YelpChi} and the testing set $\mathcal{T}_{test}$ = {Facebook, Weibo, Reddit, Amazon}. Additionally, a set of auxiliary graphs $\mathcal{T}_{aux}$ = {Flickr, CiteSeer, BlogCatalog} is used for model enhancement through either pattern extraction or finetuning.

**Case Study 1: Pattern Augmentation without Finetuning.** In this setting, we assess whether OWLEYE can improve performance by simply extracting patterns from new graphs without fine-tuning the model. We extract attribute- and structure-level patterns from graphs in $\mathcal{T}_{aux}$ and incorporate them into the dictionaries.

Table 3: Case Study 1: Exploration of OWLEYE's continuous learning capability **without finetuning**. $|\mathcal{T}_{aux}| = n$ indicates that patterns from $n$ graphs are added to the dictionaries.

| Size of $\mathcal{T}_{aux}$ | Facebook | Weibo | Reddit | Amazon | Average |
|---|---|---|---|---|---|
| $|\mathcal{T}_{aux}| = 0$ | 6.72±1.63 | 59.63±0.96 | 3.93±0.08 | 54.88±4.37 | 31.29±1.76 |
| $|\mathcal{T}_{aux}| = 1$ | 6.32±1.25 | 59.79±0.86 | 3.96±0.13 | 55.78±3.29 | 31.46±1.31 |
| $|\mathcal{T}_{aux}| = 2$ | 6.52±1.47 | 59.82±1.10 | **4.05±0.12** | 56.73±2.62 | 31.78±1.32 |
| $|\mathcal{T}_{aux}| = 3$ | **7.04±1.51** | **60.06±0.96** | 3.98±0.06 | **58.01±3.42** | **32.27±1.49** |

The results are shown in Table 3, where $|\mathcal{T}_{aux}| = n$ indicates that $n$ additional graphs are used for pattern extraction and when $|\mathcal{T}_{aux}| = 0$, no new patterns are added. We observe a consistent performance improvement as more patterns are incorporated, validating OWLEYE's ability to incrementally learn from new data sources in a plug-and-play fashion.

**Case Study 2: Pattern Augmentation with Finetuning.** Next, we investigate OWLEYE's continual learning performance when finetuning is permitted using the graphs in $\mathcal{T}_{aux}$. Table 4 presents the results under varying the number of graphs in $\mathcal{T}_{aux}$. We find that OWLEYE achieves its highest average performance when finetuned on two graphs from $\mathcal{T}_{aux}$.

Table 4: Case Study 2: Investigation of OWLEYE's continual learning performance when **finetuning is permitted** using $n$ graphs in $\mathcal{T}_{aux}$. $|\mathcal{T}_{aux}| = 0$ means that we do not finetune the model.

| Size of $\mathcal{T}_{aux}$ | Facebook | Weibo | Reddit | Amazon | Average |
|---|---|---|---|---|---|
| $|\mathcal{T}_{aux}| = 0$ | 6.72±1.63 | **59.63±0.96** | 3.93±0.08 | 54.88±4.37 | 31.29±1.76 |
| $|\mathcal{T}_{aux}| = 1$ | 6.82±1.25 | 59.39±0.78 | 3.91±0.14 | 55.06±2.86 | 31.30±1.28 |
| $|\mathcal{T}_{aux}| = 2$ | **6.93±1.89** | 59.35±0.55 | 3.93±0.13 | 55.74±3.61 | **31.48±1.54** |
| $|\mathcal{T}_{aux}| = 3$ | 6.69±1.42 | 58.12±1.42 | **4.01±0.07** | **56.53±5.34** | 31.33±2.06 |

However, a comparison with the results from Case Study 1 reveals a notable insight: OWLEYE performs better without any fine-tuning simply by leveraging the added patterns. This highlights the effectiveness and practicality of our pattern-centric design for continual learning. We conjecture that the reason that finetuning the model fail to achieve better performance is that the model is hard to train on too many graphs and thus hard to converge. The visualization of the training loss vs epoch in two cases (e.g., $|\mathcal{T}_{aux}| = 0$ and $|\mathcal{T}_{aux}| = 3$) in Appendix C.3 shows that it's hard for the model to converge when $|\mathcal{T}_{aux}| = 3$.

**Case Study 3: Impact of Dictionary Size.** Finally, we analyze how the number of stored patterns (denoted as $n_{sup}$) affects performance. In the experiment, we use $\mathcal{T}_{train} = \{\text{PubMed}, \text{CiteSeer}, \text{Questions}, \text{YelpChi}\}$ for training, and let the test set be $\mathcal{T}_{test} = \{\text{Cora}, \text{Flickr}, \text{ACM}, \text{BlogCatalog}, \text{Facebook}, \text{Weibo}, \text{Reddit}, \text{Amazon}\}$. Table 5 reports the results. We observe that: (1) Increasing the number of patterns from 10 to 200 leads to a 0.55% performance gain; and (2) Beyond 200, the improvement becomes marginal and the performance becomes stable even if we keep increasing the dictionary size. These findings demonstrate that OWLEYE benefits from a larger pattern dictionary up to a saturation point, beyond which rewards diminish.

Table 5: Case Study 3: AUPRC (%) under different dictionary sizes (i.e., $n_{sup}$).

| $n_{sup}$ | Cora | Flickr | ACM | BlogCatalog | Facebook | Weibo | Reddit | Amazon | Average |
|---|---|---|---|---|---|---|---|---|---|
| 10 | 44.55±0.48 | 37.86±0.26 | 39.57±0.08 | 34.84±0.33 | 5.63±0.73 | 61.30±0.44 | 4.03±0.12 | 55.89±2.16 | 35.46±0.58 |
| 100 | 43.52±0.78 | 37.45±0.23 | 39.87±0.06 | 34.76±0.38 | 5.43±1.34 | 61.13±0.67 | 4.03±0.08 | 60.49±2.82 | 35.84±0.79 |
| 200 | 43.24±0.50 | 37.65±0.29 | 39.92±0.12 | 34.72±0.35 | 5.32±1.40 | 60.70±0.53 | 4.02±0.08 | 62.51±1.45 | 36.01±0.59 |
| 500 | 43.28±0.92 | 37.96±0.39 | 39.86±0.13 | 34.79±0.35 | 5.43±1.32 | 61.00±0.47 | 4.04±0.11 | 62.77±1.36 | 36.14±0.63 |
| 1000 | 43.88±0.73 | 37.99±0.33 | 39.82±0.08 | 34.92±0.39 | 5.33±1.29 | 60.52±0.79 | 4.03±0.12 | 61.76±2.50 | 36.03±0.78 |
| 2000 | 43.94±0.46 | 37.69±0.25 | 39.75±0.13 | 34.99±0.31 | 5.62±1.17 | 60.90±0.21 | 4.25±0.11 | 62.20±3.18 | **36.17±0.73** |

## 4 RELATED WORK

Graph anomaly detection (GAD) is widely used in many applications (Grubbs, 1969; Ma et al., 2021; Pourhabibi et al., 2020; Li et al., 2021; Duan et al., 2023) that naturally involve graph-structured data, such as transaction networks in financial fraud detection (Slipenchuk & Epishkina, 2019; Ramakrishnan et al., 2019), communication and access networks in cybersecurity intrusion detection (Brdiczka et al., 2012; Duan et al., 2023), and user-user interaction graphs in fake news detection on social networks (Shu et al., 2017; 2019; Fu & He, 2021). In recent years, the success of deep learning has spurred growing interest in developing deep learning-based GAD methods (Ma et al., 2023). Depending on the availability of labeled data, deep GAD approaches can be broadly categorized into supervised and unsupervised methods (Qiao et al., 2024). In the unsupervised setting, graph contrastive learning methods (Zheng et al., 2021b; Liu et al., 2022; Chen et al., 2022; Jin et al., 2021; Xu et al., 2022) aim to learn effective node or graph-level representations by pulling similar instances together in the embedding space without any label information. Alternatively, reconstruction-based methods (Ding et al., 2019; Huang et al., 2023; Li et al., 2019; Luo et al., 2022; Peng et al., 2023) focus on learning low-dimensional embeddings capable of reconstructing input graph attributes or structures, with anomalies identified as instances exhibiting high reconstruction errors. In the supervised setting, generative GNN-based methods leverage label information to augment training data by synthesizing high-quality graph signals. Representative works include GraphSMOTE (Zhao et al., 2021), GraphMixup (Wu et al., 2022), and GraphENS (Park et al., 2022), which enhance model generalization and robustness. More recently, the advent of large language models (LLMs) has sparked a paradigm shift in AI research due to their strong generalization capabilities. Motivated by this, researchers are exploring "one-for-all" generalist frameworks (Niu et al., 2024; Liu et al., 2024) capable of adapting to diverse, unseen graph domains with minimal task-specific tuning. Cross-domain graph anomaly detection (CD-GAD) has recently drawn growing interest as models trained on one graph often degrade when deployed on graphs with different structures or feature distributions (Ding et al., 2021; Pirhayatifard & Silva). Wang et al. (2023) proposed an anomaly-aware contrastive alignment approach that explicitly incorporates anomaly signals into cross-domain representation learning, improving robustness under heterogeneous domains. This paper also aims to develop such a generalist framework for GAD, while addressing several open challenges in existing approaches, including inadequate graph alignment across domains, lack of continual learning capabilities, and poor performance in zero-shot anomaly detection scenarios.

## 5 CONCLUSION

In this work, we presented OWLEYE, a novel generalist framework for zero-shot graph anomaly detection across multiple domains. To address the limitations of existing methods, such as poor feature alignment, lack of continual evolving capabilities, and reliance on labeled target data, OWL-EYE introduces a structured and interpretable solution. By storing representative normal patterns in a reusable dictionary, OWLEYE enables scalable and effective anomaly detection on entirely unseen graphs without retraining or target supervision. Extensive experiments on real-world datasets validate the superior performance and transferability of OWLEYE over existing state-of-the-art approaches.

ACKNOWLEDGMENT

This work is supported by National Science Foundation under Award No. IIS-2117902. The views and conclusions are those of the authors and should not be interpreted as representing the official policies of the funding agencies or the government.

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

## A    IMPLEMENTATION DETAILS

The neural network structure of the proposed framework is 3-layer GCN. In the experiments, we set the initial learning rate to be 3e-5, the hidden feature dimension to be 512 and use Adam as the optimizer. The experiments are performed on a Windows machine with a 24GB RTX 4090 GPU.

## B    ADDITIONAL EXPERIMENTAL RESULTS

### B.1    ADDITIONAL EFFECTIVENESS ANALYSIS

In the experiment, we evaluate our method in the zero-shot and 10-shot setting on the graph from the test set $\mathcal{T}_{test}$. For the supervised learning methods (e.g., BWGNN and GHRN), we evaluate these two methods in the 10-shot setting, where we randomly sample 5 normal nodes and 5 anomalies as the support nodes. Table 6 and Table 7 show the performance of OWLEYE and the baseline methods with respect to AUROC in the zero-shot and 10-shot setting, respectively. OWLEYE demonstrates strong anomaly detection capability in the generalist GAD scenario, without any finetuning. The average AUPRC is better than the best competitor (i.e., GHRN).

Table 6: Anomaly detection performance in zero-shot setting w.r.t AUROC. We highlighted the results ranked first and second. "Average" indicates the average AUROC over 8 datasets.

| Method | Cora | Flickr | ACM | BlogCatalog | Facebook | Weibo | Reddit | Amazon | Average |
|---|---|---|---|---|---|---|---|---|---|
| | | | | Supervised (10-shot) | | | | | |
| BWGNN | 60.53±7.76 | 69.40±9.22 | 70.37±13.29 | 68.74±6.80 | 67.74±11.28 | 61.28±2.58 | 63.68±8.21 | 69.75±5.81 | 66.44±7.49 |
| GHRN | 76.59±3.55 | 71.96±4.19 | 76.83±4.35 | 72.73±4.72 | 75.98±2.99 | 82.79±4.15 | 69.76±5.52 | 74.35±4.49 | 75.12±4.25 |
| | | | | Unsupervised (zero-shot) | | | | | |
| DOMINANT | 85.75±1.43 | 73.43±3.12 | 73.62±1.26 | 70.27±2.37 | 52.31±1.39 | 85.42±12.64 | 53.49±2.94 | 64.73±2.73 | 69.88±3.49 |
| SLGAD | 72.73±0.70 | 63.07±1.58 | 50.88±2.03 | 60.36±2.19 | 45.57±2.51 | 60.16±2.50 | 50.28±4.48 | 54.63±1.25 | 57.21±2.15 |
| TAM | 57.85±0.95 | 62.39±1.54 | 75.16±2.68 | 62.39±3.52 | 63.30±0.12 | 71.33±0.07 | 52.37± 0.21 | 76.04±3.62 | 65.10±1.58 |
| CARE | 66.92±0.62 | 67.90±0.70 | 70.95±0.75 | 45.41±0.59 | 79.18±1.69 | 86.34±0.14 | 52.51±0.28 | 83.09±0.65 | 69.26±0.65 |
| ARC | 84.92±0.58 | 72.34±0.16 | 77.54±0.13 | 73.65±0.55 | 65.03±0.79 | 88.43±0.62 | 58.46±2.20 | 71.93±3.84 | 74.04±1.11 |
| UNPrompt | 63.55±3.09 | 69.67±0.47 | 71.98±1.38 | 67.83±2.73 | 63.87±7.37 | 47.44±2.77 | 55.01±1.20 | 56.77±6.73 | 62.01±3.22 |
| OWLEYE | 79.52±0.43 | 74.81±0.97 | 78.20±0.12 | 74.83±0.26 | 64.85±0.37 | 87.66±0.14 | 58.03±0.68 | 85.43±0.79 | 75.42±0.47 |

Table 7: Anomaly detection performance in 10-shot setting w.r.t AUROC. We highlighted the results ranked first and second. "Average" indicates the average AUROC over 8 datasets.

| Method | Cora | Flickr | ACM | BlogCatalog | Facebook | Weibo | Reddit | Amazon | Average |
|---|---|---|---|---|---|---|---|---|---|
| | | | | Supervised (10-shot) | | | | | |
| BWGNN | 60.53±7.76 | 69.40±9.22 | 70.37±13.29 | 68.74±6.80 | 67.74±11.28 | 61.28±2.58 | 63.68±8.21 | 69.75±5.81 | 66.44±7.49 |
| GHRN | 76.59±3.55 | 71.96±4.19 | 76.83±4.35 | 72.73±4.72 | 75.98±2.99 | 82.79±4.15 | 69.76±5.52 | 74.35±4.49 | 75.12±4.25 |
| | | | | Unsupervised & Finetune (10-shot) | | | | | |
| DOMINANT | 73.52±0.47 | 73.84±2.76 | 73.82±0.19 | 74.39±0.10 | 51.22±0.85 | 91.66±0.27 | 53.48±4.87 | 60.39±2.80 | 69.04±1.54 |
| SLGAD | 73.39±0.84 | 64.15±1.33 | 53.55±1.82 | 62.67±1.84 | 51.40±2.51 | 61.64±1.91 | 53.17±3.95 | 55.23±1.36 | 59.40±1.94 |
| TAM | 62.56±2.10 | 65.19±1.86 | 86.29±1.57 | 63.69±0.88 | 76.26±3.70 | 71.73±0.16 | 56.62±0.49 | 77.13±4.62 | 69.93±1.92 |
| CARE | 67.27±2.44 | 68.81±0.76 | 71.53±0.71 | 53.95±1.99 | 79.94±10.28 | 87.65±3.08 | 55.32±0.48 | 83.80±3.54 | 71.03±2.91 |
| ARC | 85.28±0.38 | 74.62±0.28 | 77.84±0.17 | 73.58±0.33 | 67.28±1.18 | 87.04±1.36 | 60.06±1.21 | 73.79±3.16 | 74.94±1.01 |
| UNPrompt | 65.06±0.41 | 69.16±0.47 | 73.24±0.45 | 68.95±0.25 | 67.13±4.01 | 53.21±2.16 | 55.69±0.96 | 62.14±0.97 | 64.32±1.21 |
| OWLEYE | 79.71±0.72 | 75.77±1.36 | 78.06±0.34 | 74.42±0.33 | 66.94±0.50 | 88.61±0.36 | 57.82±0.75 | 84.93±2.82 | 75.78±0.90 |

### B.2    PERFORMANCE EVALUATION WITH DIFFERENT DATASET SPLIT

In the experiments, we evaluate OWLEYE and the baseline methods on different training and test set split. Specifically, the training datasets $\mathcal{T}_{train}$ consist of Reddit, BlogCatalog, Questions, and Cora, while the testing datasets $\mathcal{T}_{test}$ consist of Flickr, CiteSeer, PubMed, ACM, Facebook, Weibo, YelpChi, and Amazon. The experimental results are shown in Table 8 and Table 9. OWLEYE demonstrates strong anomaly detection capability in the generalist GAD scenario, without any fine-tuning. Specifically, OWLEYE achieves state-ofthe-art performance on 4 out of 8 datasets in terms of AUROC and 3 out of 8 in terms of AURPC and demonstrates competitive performance on the remainder.

### B.3    CAN CROSS-DOMAIN FEATURE ALIGNMENT BENEFIT OTHER METHOD?

In this subsection, we want to answer the following question: *Can cross-domain feature alignment benefit other method?* We conducted experiments by replacing ARC's feature preprocessing with

Table 8: Anomaly detection performance in zero-shot setting w.r.t AUPRC. We highlighted the results ranked first and second. "Average" indicates the average AUPRC over 8 datasets.

| Method | Flickr | CiteSeer | PubMed | ACM | Weibo | Facebook | YelpChi | Amazon | Average |
|---|---|---|---|---|---|---|---|---|---|
| | | | | Supervised (10-shot) | | | | | |
| BWGNN | 12.39±2.68 | 10.31±1.99 | 11.63±2.87 | 13.37±6.03 | 9.55±2.12 | 5.81±1.17 | 2.45±3.76 | 12.40±1.86 | 9.74±2.81 |
| GHRN | 16.45±2.59 | 14.04±0.73 | 16.45±2.59 | 16.29±1.41 | 17.51±1.52 | 6.24±1.12 | 6.29±1.41 | 13.84±2.63 | 13.39±1.75 |
| | | | | Unsupervised (zero-shot) | | | | | |
| DOMINANT | 28.76±1.52 | 18.74±2.71 | 14.32±1.66 | 32.49±4.97 | 29.63±4.98 | 3.42±0.86 | 4.73±0.65 | 36.80±8.37 | 21.11±3.21 |
| SLGAD | 16.93±8.20 | 3.87±0.83 | 11.07±2.05 | 1.33±0.23 | 35.80±1.41 | 0.93±0.23 | 5.60±1.44 | 5.33±2.20 | 10.11±2.08 |
| TAM | 23.34±1.42 | 8.80±0.88 | 23.71±1.22 | 40.68±2.58 | 23.01±15.14 | 4.18±1.42 | 6.19±0.71 | 45.26±4.34 | 22.02±3.47 |
| CARE | 35.64±0.16 | 12.47±0.72 | 21.62±1.29 | 37.76±0.35 | 40.70±0.74 | 5.52±0.34 | 6.51±0.66 | 56.76±1.44 | 26.00±0.71 |
| ARC | 36.72±0.14 | 45.94±0.39 | 28.61±0.15 | 39.26±0.20 | 63.05±0.75 | 5.39±0.45 | 5.26±0.13 | 27.47±9.59 | 31.46±1.47 |
| UNPrompt | 24.35±0.41 | 5.81±0.55 | 10.14±0.45 | 14.89±0.67 | 33.43±6.55 | 3.31±1.16 | 7.99±2.25 | 9.74±0.57 | 13.71±1.58 |
| OWLEYE | 37.69±0.15 | 43.14±1.23 | 29.72±0.43 | 39.83±0.19 | 59.30±1.88 | 5.45±0.59 | 6.58±0.17 | 60.46±6.47 | 35.27±1.39 |

Table 9: Anomaly detection performance in zero-shot setting w.r.t AUROC. We highlighted the results ranked first and second. "Average" indicates the average AUROC over 8 datasets.

| Method | Flickr | CiteSeer | PubMed | ACM | Weibo | Facebook | YelpChi | Amazon | Average |
|---|---|---|---|---|---|---|---|---|---|
| | | | | Supervised (10-shots) | | | | | |
| BWGNN | 69.40±9.22 | 63.05±3.63 | 64.16±5.49 | 70.37±13.29 | 61.28±2.58 | 67.74±11.28 | 54.22±3.41 | 69.75±5.81 | 65.00±6.84 |
| GHRN | 71.96±4.19 | 77.59±3.55 | 76.96±4.19 | 76.83±4.35 | 82.79±4.15 | 75.98±2.99 | 53.83±4.35 | 74.35±4.49 | 73.79±3.78 |
| | | | | Unsupervised (zero-shot) | | | | | |
| DOMINANT | 73.43±3.12 | 74.96±2.87 | 75.13±1.23 | 73.62±1.26 | 85.42±12.64 | 52.31±1.39 | 53.22±0.98 | 64.73±2.73 | 69.13±3.28 |
| SLGAD | 63.07±1.58 | 55.12±2.44 | 56.51±1.02 | 50.88±2.03 | 60.16±0.25 | 45.57±2.51 | 47.56±1.06 | 54.63±1.25 | 54.81±1.80 |
| TAM | 62.39±1.54 | 67.29±1.38 | 77.84±0.98 | 75.16±2.68 | 71.33±0.07 | 63.30±0.09 | 52.40±0.52 | 76.04±3.62 | 68.22±1.35 |
| CARE | 67.90±0.70 | 69.43±0.87 | 66.55±0.97 | 70.95±0.75 | 86.34±0.14 | 79.18±1.69 | 53.89±0.82 | 83.09±0.42 | 72.42±0.80 |
| ARC | 74.12±0.65 | 89.58±0.30 | 82.25±0.18 | 78.28±0.32 | 87.15±0.66 | 66.39±1.61 | 54.13±0.26 | 72.54±6.78 | 75.80±1.34 |
| UNPrompt | 69.15±0.25 | 57.70±0.52 | 71.47±1.49 | 72.68±0.33 | 66.94±5.02 | 55.05±2.11 | 60.49±5.54 | 66.84±2.06 | 65.04±2.16 |
| OWLEYE | 75.30±0.98 | 84.47±0.58 | 77.98±0.25 | 78.57±0.11 | 87.49±0.57 | 66.25±0.72 | 54.73±0.63 | 85.56±2.32 | 76.29±0.77 |

our proposed cross-domain feature alignment method. The results are shown in Table 10. The results demonstrate improved performance across 6 out of 8 datasets, achieving an average increase of 0.57 in AUCROC and 1.01 in AUPRC. This validates the effectiveness of our proposed cross domain feature alignment.

Table 10: Effectiveness of Cross-domain Feature Alignment (FA)

| Method | Cora | Flickr | ACM | BlogCatalog | Facebook | Weibo | Reddit | Amazon | Average |
|---|---|---|---|---|---|---|---|---|---|
| | | | | AUROC | | | | | |
| ARC | 84.92 ± 0.58 | 72.34 ± 0.16 | 77.54 ± 0.13 | 74.65 ± 0.55 | 65.03 ± 0.79 | 88.43 ± 0.62 | 58.46 ± 2.20 | 71.93 ± 3.84 | 74.04 ± 1.11 |
| ARC + FA | 84.06 ± 0.31 | 73.37 ± 0.21 | 77.84 ± 0.23 | 74.92 ± 0.21 | 64.93 ± 0.55 | 88.81 ± 0.75 | 59.17 ± 0.72 | 74.74 ± 4.51 | 74.61 ± 0.94 |
| Improvement | -0.86 | 1.03 | 0.30 | 0.27 | -0.10 | 0.38 | 0.71 | 2.81 | 0.57 |
| | | | | AUPRC | | | | | |
| ARC | 45.20 ± 1.08 | 35.13 ± 0.20 | 39.02 ± 0.08 | 33.43 ± 0.15 | 4.25 ± 0.47 | 64.18 ± 0.68 | 4.20 ± 0.25 | 20.48 ± 6.89 | 30.74 ± 1.23 |
| ARC + FA | 46.71 ± 0.72 | 37.15 ± 0.18 | 39.84 ± 0.22 | 34.74 ± 0.28 | 4.62 ± 0.52 | 61.79 ± 0.37 | 4.05 ± 0.37 | 25.05 ± 6.95 | 31.74 ± 1.20 |
| Improvement | 1.51 | 2.02 | 0.82 | 1.31 | 0.37 | -2.39 | -0.15 | 4.57 | 1.01 |

## B.4 EXPERIMENTS ON HIGH ANOMALY RATE DATASET

In this subsection, we control Facebook data with synthetic anomaly rate. The raw anomaly rate of Facebook is 2.31% and we manually inject 10% and 20% more anomalies. The results are shown in Table 11. The results (AUPRC) show that our method consistently outperforms two baseline methods when the anomalies rate increases.

## B.5 EXPERIMENT RESULTS ABOUT SIGNIFICANT DOMAIN SHIFT

In this subsection, we evaluate our method under a heavy domain shift setting. To simulate a scenario where the e-commerce co-review domain is entirely unseen during training, we remove the YelpChi graph from the training set. This creates a substantial distribution mismatch, as no graph from the e-commerce co-review domain is available during model learning. In the test phase, we examine the effect of removing YelpChi graph on Amazon graph as both of them are from the same domain (e-commerce co-review domain), while the rest 10 graphs from other domains.

In Table 12, the results reveal that, under this heavy domain shift, the performance on the Amazon graph decreases by 4.6% in AUPRC, while its AUROC remains largely unchanged. This behavior is expected: with an entire domain missing during training, domain-specific patterns become harder to recover. Nevertheless, the performance drop remains moderate, demonstrating that the patterns extracted from other training graphs still help support robust anomaly detection on the Amazon

Table 11: Performance of OWLEYE w.r.t AUPRC with different anomaly rates

| Anomaly Rate | 2.31% + 0% | 2.31% + 10% | 2.31% + 20% |
|---|---|---|---|
| UNPrompt | 4.32±0.55 | 53.38±0.56 | 70.09±0.28 |
| ARC | 4.25 ±0.47 | 54.02 ±0.41 | 71.92 ±0.84 |
| OWLEYE | **4.86 ±0.33** | **54.47 ±0.82** | **72.47±0.48** |

dataset. Interestingly, for graphs from other domains, we observe slight improvements in AUROC and AUPRC when YelpChi is removed. For example, Cora's AUROC increases from 0.7952 to 0.7989 and AUPRC from 0.4394 to 0.4476, while Flickr's AUROC rises from 0.7481 to 0.7636. This trend is consistent across several other non-e-commerce graphs, suggesting that removing one domain might slightly reduce its influence during training, thereby allowing the model to better capture patterns from other domains. Overall, these findings indicate that while heavy domain shift does lead to reasonable performance degradation for the affected domain, the cross-domain structural and attribute patterns captured by our model continue to provide meaningful generalization.

Table 12: Domain Shift Experiment Results (in %), testing on the Amazon graph. We remove the YelpChi graph from the e-commerce co-review domain to simulate significant domain shifts.

| Dataset | With YelpChi | | Without YelpChi | |
|---|---|---|---|---|
| | **AUROC (%)** | **AUPRC (%)** | **AUROC (%)** | **AUPRC (%)** |
| Cora | 79.52 ± 0.43 | 43.94 ± 0.46 | 79.89 ± 0.27 | 44.76 ± 0.51 |
| Flickr | 74.81 ± 0.97 | 37.69 ± 0.25 | 76.36 ± 2.55 | 37.91 ± 0.38 |
| ACM | 78.20 ± 0.12 | 39.75 ± 0.13 | 78.93 ± 0.87 | 39.72 ± 0.15 |
| BlogCatalog | 74.83 ± 0.26 | 34.99 ± 0.31 | 75.63 ± 1.46 | 35.04 ± 0.36 |
| Facebook | 64.85 ± 0.37 | 5.62 ± 1.17 | 66.27 ± 0.97 | 6.19 ± 1.37 |
| Weibo | 87.65 ± 0.14 | 60.90 ± 0.21 | 88.10 ± 0.22 | 61.18 ± 1.04 |
| Reddit | 58.02 ± 0.68 | 4.25 ± 0.11 | 57.79 ± 0.22 | 3.97 ± 0.06 |
| Amazon | 85.43 ± 0.79 | 62.20 ± 3.18 | 85.15 ± 0.80 | 57.64 ± 3.75 |

## B.6 EFFECTIVENESS OF STRUCTURAL REPRESENTATION

In Figure 3 (OWLEYE vs OWLEYE-S), we have validated that including both attribute-level representation and structural-level representation indeed help successfully identify more anomalies. In this subsection, we further verify the necessity of including structural representation in our model design. Table 13 shows the experimental results comparing using both structural similarity and attribute similarity (A+S) for domain similarity measurement vs only using structural similarity (S-Only) for domain similarity measurement. The experimental results show that using both structural similarity and attribute similarity (A+S) for domain similarity measurement decreases the performance. This suggests that using both structural and attribute similarity for domain similarity measurement is less stable than relying on structural similarity alone, because camouflaged anomalies may mimic normal neighbors' attributes and cross-domain feature discrepancies make reliable measurement more challenging. Table 14 shows that including structural patterns indeed increases the performance.

## B.7 EFFECTIVENESS OF DIFFERENT FEATURE REDUCTION METHODS

In this subsection, we compared different linear and nonlinear feature projection methods including PCA, SVD, Kernel PCA, and NMF in Table 15. The experimental results show that our method with PCA still achieves the best performance as modeled in this paper. Comparing PCA with nonlinear methods like Kernel PCA and NMF, we observe that using more complicated feature projection does not necessarily improve the performance, as it might distort and misalign the original feature space, leading to performance drop. When the feature dimension is smaller than the preset projection dimension, we use Gaussian Random Projection to 256 following ARC and then do the feature reduction.

Table 13: Comparison of using both attribute and structural similarity (A+S) vs structural similarity only (S-Only).

| Dataset | A+S (%) | S-Only (%) | Improvement (%) |
|---|---|---|---|
| Cora | 43.26 ± 0.54 | 43.94 ± 0.46 | 0.68 |
| Flickr | 37.83 ± 0.39 | 37.69 ± 0.25 | -0.14 |
| ACM | 39.84 ± 0.28 | 39.75 ± 0.13 | -0.09 |
| BlogCatalog | 34.34 ± 0.53 | 34.99 ± 0.31 | 0.65 |
| Facebook | 6.11 ± 1.35 | 5.62 ± 1.17 | -0.49 |
| Weibo | 58.61 ± 5.18 | 60.90 ± 0.21 | 2.28 |
| Reddit | 4.05 ± 0.12 | 4.25 ± 0.11 | 0.20 |
| Amazon | 48.01 ± 18.44 | 62.20 ± 3.18 | 14.19 |
| Average | 34.26 ± 3.35 | 36.17 ± 0.73 | 2.16 |

Table 14: Comparison of using No Structural Patterns vs OwlEye for AUPRC.

| Dataset | No Structural Patterns | OwlEye | Improvement (%) |
|---|---|---|---|
| Cora | 39.38 ± 1.32 | 43.94 ± 0.46 | 4.56 |
| Flickr | 38.06 ± 0.08 | 37.69 ± 0.25 | -0.36 |
| ACM | 39.64 ± 0.15 | 39.75 ± 0.13 | 0.11 |
| BlogCatalog | 35.42 ± 0.26 | 34.99 ± 0.31 | -0.43 |
| Facebook | 4.71 ± 0.27 | 5.62 ± 1.17 | 0.90 |
| Weibo | 57.18 ± 1.09 | 60.90 ± 0.21 | 3.72 |
| Reddit | 4.12 ± 0.15 | 4.25 ± 0.11 | 0.13 |
| Amazon | 61.98 ± 1.08 | 62.20 ± 3.18 | 0.22 |
| Average | 35.06 ± 0.55 | 36.17 ± 0.73 | 1.11 |

## B.8 PATTERNS FROM ONE GRAPH VS MULTIPLE GRAPHS

In this subsection, we aim to investigate whether incorporating patterns from multiple graphs can benefit anomaly detection. The results in Table 16 indicate that leveraging patterns from multiple graphs consistently improves performance compared to using patterns from a single target graph. Specifically, the average AUPRC across eight graphs increases by 0.32%, demonstrating that cross-graph knowledge can provide complementary information that helps identify anomalies more effectively. We observe that datasets such as Cora, Weibo, and Amazon benefit the most, with improvements of 0.73%, 0.43%, and 0.48%, respectively, suggesting that in domains with diverse structures or large graphs, shared patterns from multiple sources are particularly valuable. A few datasets, like ACM, show minor negative change (-0.09%), which may be attributed to the already sufficient patterns present in the target graph, highlighting that the benefit of cross-graph patterns depends on the intrinsic complexity and variability of the graph. Overall, these results empirically validate that maintaining a structured dictionary with patterns from multiple graphs enhances the model's generalization capability for detecting anomalies in unseen domains.

## B.9 MEDIAN OPERATION VS MEAN OPERATION IN FEATURE NORMALIZATION

In Table 17, we compare the use of median and mean operations in our feature preprocessing module. (Table 19 shows the norm of raw features in each graph.) The results show that replacing the median with the mean significantly reduces performance on Cora ($43.63 \rightarrow 34.95$), CiteSeer ($42.57 \rightarrow 33.65$), and Amazon ($56.07 \rightarrow 38.56$), indicating that the mean is sensitive to graphs with unusually large feature norms, which can dominate the normalization process and distort the shared feature space. In contrast, median aggregation preserves the structural and semantic information across graphs, maintaining more stable performance on all datasets (e.g., Flickr: 37.92 vs 37.64, ACM: 39.55 vs 39.05). This demonstrates that using the median is more robust for heterogeneous graph domains, especially when feature distributions vary significantly across graphs.

Table 15: Comparison of different feature preprocessing methods (AUPRC %).

| Dataset | PCA | SVD | Kernel PCA | NMF |
|---|---|---|---|---|
| Cora | 43.94 ± 0.46 | 44.13 ± 0.81 | 44.05 ± 0.68 | 15.06 ± 2.28 |
| Flickr | 37.69 ± 0.25 | 38.18 ± 0.37 | 37.54 ± 0.46 | 33.09 ± 0.70 |
| ACM | 39.75 ± 0.13 | 39.18 ± 0.22 | 38.75 ± 0.15 | 32.28 ± 1.13 |
| BlogCatalog | 34.99 ± 0.31 | 35.28 ± 0.31 | 34.93 ± 0.24 | 33.36 ± 0.48 |
| Facebook | 5.62 ± 1.17 | 5.00 ± 0.27 | 5.63 ± 1.05 | 7.43 ± 1.34 |
| Weibo | 60.90 ± 0.21 | 57.89 ± 2.48 | 60.55 ± 0.22 | 49.90 ± 3.66 |
| Reddit | 4.25 ± 0.11 | 3.44 ± 0.24 | 4.10 ± 0.15 | 3.42 ± 0.07 |
| Amazon | 62.20 ± 3.18 | 44.27 ± 3.45 | 38.75 ± 3.07 | 20.61 ± 7.57 |
| Average | **36.17 ± 0.73** | 33.42 ± 1.02 | 33.04 ± 0.75 | 24.39 ± 2.15 |

Table 16: Comparison of using patterns from the target graph only versus patterns from multiple graphs (AUPRC %).

| Dataset | Patterns from target graph only | Patterns from multiple graphs | Improvement |
|---|---|---|---|
| Cora | 43.21 ± 0.98 | 43.94 ± 0.46 | 0.73 |
| Flickr | 37.41 ± 0.16 | 37.69 ± 0.25 | 0.28 |
| ACM | 39.84 ± 0.23 | 39.75 ± 0.13 | -0.09 |
| BlogCatalog | 34.81 ± 0.44 | 34.99 ± 0.31 | 0.17 |
| Facebook | 5.27 ± 1.34 | 5.62 ± 1.17 | 0.35 |
| Weibo | 60.47 ± 0.75 | 60.90 ± 0.21 | 0.43 |
| Reddit | 4.03 ± 0.08 | 4.25 ± 0.11 | 0.23 |
| Amazon | 61.72 ± 3.01 | 62.20 ± 3.18 | 0.48 |
| **Average** | 35.84 ± 0.88 | **36.17 ± 0.73** | 0.32 |

### B.10 ATYPICAL TRAINING DOMAIN EXPERIMENT

In Table 18, we examine the robustness of our method when a training domain is highly atypical by varying the training graphs. Since Weibo has the largest feature norm among all graphs shown in Table 19, we conduct two experiments with different training sets:

- 1) With Weibo: Pubmed, Questions, Weibo, and YelpChi;
- 2) With CiteSeer: Pubmed, Questions, CiteSeer, and YelpChi.

As shown in Table 18, the results indicate that most datasets maintain consistent performance across the two settings. For instance, Cora achieves 43.63% AUPRC with Weibo and 43.94% with CiteSeer, ACM scores 39.55% versus 39.75%, and BlogCatalog shows 35.00% versus 34.99%. The average performance is slightly higher when CiteSeer is included (32.63% vs. 31.56%), suggesting that our method is robust to the choice of a highly atypical domain in the training set.

### B.11 MORE EFFICIENCY ANALYSIS RESULTS

In Table 20, we report the training time on the ACM dataset for all baseline methods in the table. Following the experiment on efficiency analysis shown in Figure 3, ACM is selected in the experiment for the better comparison between training time and fine-tuning time. The results show that the training time of our method is more efficient than most of the baseline methods with overall first place performance gain reported in Tables 1 and 2.

### B.12 PARAMETER SENSITIVITY ANALYSIS

In this section, we conduct a comprehensive sensitivity analysis of the key hyperparameters on eight datasets, with the results summarized in Table 21. We focus on four hyperparameters: (1) the constant positive temperature $\tau \in \{1, 2, 3, 4\}$, which controls the degree of feature normaliza-

Table 17: AUPRC scores comparing median vs. mean aggregation.

| Dataset | Median (AUPRC %) | Mean (AUPRC %) |
|---|---|---|
| Cora | 43.63 ± 1.94 | 34.95 ± 11.96 |
| Flickr | 37.92 ± 0.42 | 37.64 ± 1.15 |
| ACM | 39.55 ± 0.12 | 39.05 ± 0.31 |
| BlogCatalog | 35.00 ± 0.20 | 34.53 ± 0.63 |
| Facebook | 4.56 ± 0.71 | 4.30 ± 1.15 |
| CiteSeer | 42.57 ± 1.05 | 33.65 ± 12.64 |
| Reddit | 4.19 ± 0.14 | 3.98 ± 0.16 |
| Amazon | 56.07 ± 2.28 | 38.56 ± 17.31 |
| **Average** | 32.94 ± 0.86 | 28.33 ± 5.66 |

Table 18: AUPRC scores when including patterns from Weibo vs. CiteSeer.

| Dataset | With Weibo (AUPRC %) | With CiteSeer (AUPRC %) |
|---|---|---|
| Cora | 43.63 ± 1.94 | 43.94 ± 0.46 |
| Flickr | 37.92 ± 0.42 | 37.69 ± 0.25 |
| ACM | 39.55 ± 0.12 | 39.75 ± 0.13 |
| BlogCatalog | 35.00 ± 0.20 | 34.99 ± 0.31 |
| Facebook | 4.56 ± 0.71 | 5.62 ± 1.17 |
| Reddit | 4.19 ± 0.14 | 4.25 ± 0.11 |
| Amazon | 56.07 ± 2.28 | 62.20 ± 3.18 |
| **Average** | 31.56 ± 0.83 | 32.63 ± 0.80 |

tion; (2) the truncated attention scaling factor $\tau_a \in \{0.1, 0.01, 0.001, 0.0001\}$, which magnifies the significance of the selected patterns; (3) the triplet loss margin $\lambda \in \{0.1, 0.2, 0.5, 1\}$, which determines the separation strength between positive and negative pairs; and (4) the balance coefficient $\beta \in \{1, 0.1, 0.01, 0.001\}$, which controls the relative importance between graph structural features and graph attribute features.

The experimental results show that our method is highly robust to $\tau$, $\tau_a$, and $\lambda$. Specifically, across all tested values of these parameters, the performance only varies slightly within the range of $0.7502$ to $0.7552$. This indicates that the model remains stable under different choices of normalization temperature, attention scaling, and triplet loss margin. In contrast, the parameter $\beta$ has a more pronounced impact. When $\beta = 1$, the performance drops to $0.7069$, suggesting that overemphasizing one type of feature (structural feature) may degrade the model's ability to capture complementary information. As $\beta$ decreases to $0.1$, the performance improves significantly to $0.7556$, and remains consistently around $0.7550$ for smaller values. This demonstrates that a balanced contribution from both graph structural and attributed features is crucial for achieving optimal performance.

Overall, these results highlight two important findings: (i) our method exhibits strong robustness with respect to most hyperparameters, making it reliable in practical applications without heavy parameter tuning, and (ii) careful adjustment of the balance coefficient $\beta$ is particularly beneficial for enhancing performance by effectively leveraging the complementary strengths of structural and attributed graph information.

### B.13 IMPACT OF DIFFERENT VALUES OF $k$

In this subsection, we evaluate the impact of different values of $k$ in truncating the small values in truncated attention mechanism. In the experiment, $k$ is a relative value according to the dictionary size. We set the value of $k$ to $p\%$ of patterns in the dictionary, where $n_{sup} = 1000$ for each graph. We vary the percentage of it from 5% to 50% as well as two specific numbers of $k$ ($k = 5$ and $k = 10$) and report the overall results across eight graphs below. Table 22 shows the impact of $k$ in the truncated attention mechanism.

Table 19: Feature norms of different graphs.

| Dataset | Norm |
|---|---|
| Pubmed | 0.24 |
| Questions | 0.45 |
| Weibo | 77.82 |
| YelpChi | 0.07 |
| Cora | 2.41 |
| Flickr | 0.25 |
| ACM | 0.10 |
| BlogCatalog | 0.30 |
| Facebook | 2.38 |
| Citeseer | 2.81 |
| Reddit | 0.02 |
| Amazon | 0.49 |

Table 20: Training time comparison (in seconds) across different methods.

| Method | Training Time (s) |
|---|---|
| BWGNN | 16.6 |
| GHRN | 8.72 |
| SLGAD | 280 |
| Dominant | 470 |
| TAM | 254.71 |
| CARE | 550.23 |
| ARC | 1.89 |
| UNPrompt | 21.23 |
| OWLEYE(ours) | 3.84 |

To be more specific, the range of the value of $k$ is based on the observation that the percentage of the anomaly is usually lower than 10%. Using the ratio of dictionary size can filter out most of the anomalies and then reduce the uncertainty. In a situation where the percentage of anomalies is unusually high, we can manually increase the value of $k$. The results show that when we set the value of k to 10% to 30% of patterns in the dictionary, the AUPRC scores are around 36.1% and its performance decreases if we increase the percentage to 50%. One possible explanation is that we filter out too many patterns that might be useful for identifying anomalies. Similarly, when we decreased the value of $k$ to a value that could not filter out most of the anomalies, the performance starts to decrease, as the method involves the anomalies for feature construction.

## C VISUALIZATION

### C.1 ADDITIONAL FEATURE PREPROCESSING COMPARISON

In this subsection, we visualize the different feature preprocessing methods on Cora and Weibo datasets, showing how well different feature preprocessing methods preserve the graph structure for the graphs from two different domains. Notice that Weibo has the largest feature norm (77.82), while the norms of other graphs are less than 3. To better visualize the different data distribution in the raw graphs across different domains, Weibo is selected. We also selected Cora, as it is a citation network, different from Weibo as a social network. Similar to Figure 1, we have the similar observation in Figure 5, where ARC pushes the two graphs apart rather than aligning them in the top row and UNPrompt reverses the different patterns of Normal-Normal pairs pair and Normal-Anomaly pairs in the middle row. In contrast, our method can fairly preserve the key graph structure even if two graphs are from different domains.

Table 21: Parameter Analysis on $\tau$, $\tau_a$, $\lambda$, and $\beta$

| Parameter | Cora | Flickr | ACM | BlogCatalog | Facebook | Weibo | Reddit | Amazon | Average |
|---|---|---|---|---|---|---|---|---|---|
| $\tau = 1$ | 76.97 ± 0.73 | 73.65 ± 0.25 | 77.94 ± 0.21 | 74.04 ± 0.25 | 65.12 ± 0.81 | 87.12 ± 0.21 | 59.37 ± 0.25 | 86.22 ± 0.44 | 75.05 ± 0.39 |
| $\tau = 2$ | 80.31 ± 1.71 | 75.12 ± 1.38 | 78.20 ± 0.78 | 75.45 ± 1.44 | 64.03 ± 1.65 | 87.46 ± 0.49 | 57.73 ± 1.76 | 83.37 ± 0.89 | 75.26 ± 1.26 |
| $\tau = 3$ | 80.15 ± 1.09 | 75.94 ± 3.37 | 78.54 ± 0.63 | 75.18 ± 1.09 | 65.26 ± 2.78 | 87.40 ± 0.84 | 57.86 ± 0.99 | 81.31 ± 2.89 | 75.21 ± 1.71 |
| $\tau = 4$ | 80.30 ± 1.35 | 76.02 ± 3.46 | 78.67 ± 0.80 | 75.19 ± 1.12 | 64.99 ± 2.79 | 87.28 ± 0.70 | 57.88 ± 0.94 | 81.48 ± 5.89 | 75.23 ± 2.13 |
| $\tau_a = 0.1$ | 79.06 ± 0.26 | 75.19 ± 1.40 | 77.95 ± 0.15 | 74.77 ± 0.12 | 62.86 ± 0.52 | 87.43 ± 0.16 | 57.83 ± 0.36 | 85.33 ± 0.84 | 75.05 ± 0.48 |
| $\tau_a = 0.01$ | 79.32 ± 0.54 | 74.47 ± 0.29 | 78.17 ± 0.25 | 74.77 ± 0.13 | 63.46 ± 0.46 | 87.53 ± 0.16 | 57.88 ± 0.61 | 84.60 ± 0.83 | 75.02 ± 0.41 |
| $\tau_a = 0.001$ | 76.97 ± 0.73 | 73.65 ± 0.25 | 77.94 ± 0.21 | 74.04 ± 0.25 | 65.12 ± 0.81 | 87.12 ± 0.21 | 59.37 ± 0.25 | 86.22 ± 0.44 | 75.05 ± 0.39 |
| $\tau_a = 0.0001$ | 79.95 ± 0.56 | 75.07 ± 1.06 | 78.55 ± 0.61 | 75.11 ± 0.80 | 65.19 ± 0.73 | 87.57 ± 0.13 | 57.87 ± 1.61 | 84.85 ± 1.19 | 75.52 ± 0.84 |
| $\lambda = 0.1$ | 79.61 ± 0.60 | 75.59 ± 1.75 | 78.15 ± 0.07 | 74.84 ± 0.21 | 64.63 ± 0.38 | 87.74 ± 0.20 | 58.43 ± 0.41 | 85.14 ± 0.89 | 75.52 ± 0.56 |
| $\lambda = 0.2$ | 76.97 ± 0.73 | 73.65 ± 0.25 | 77.94 ± 0.21 | 74.04 ± 0.25 | 65.12 ± 0.81 | 87.12 ± 0.21 | 59.37 ± 0.25 | 86.22 ± 0.44 | 75.05 ± 0.39 |
| $\lambda = 0.5$ | 79.50 ± 0.52 | 75.14 ± 1.18 | 78.13 ± 0.14 | 74.78 ± 0.48 | 65.45 ± 0.56 | 87.66 ± 0.16 | 57.02 ± 0.68 | 85.35 ± 0.48 | 75.38 ± 0.53 |
| $\lambda = 1$ | 79.28 ± 0.33 | 74.77 ± 0.85 | 78.08 ± 0.16 | 74.47 ± 0.43 | 66.92 ± 0.73 | 87.44 ± 0.09 | 55.40 ± 0.73 | 85.34 ± 0.45 | 75.21 ± 0.45 |
| $\beta = 1$ | 81.54 ± 1.42 | 84.10 ± 0.18 | 85.14 ± 1.55 | 79.17 ± 0.09 | 33.60 ± 2.24 | 83.28 ± 2.38 | 52.68 ± 0.58 | 66.03 ± 2.59 | 70.69 ± 1.38 |
| $\beta = 0.1$ | 83.06 ± 1.56 | 79.87 ± 2.91 | 79.96 ± 0.53 | 76.99 ± 1.12 | 57.90 ± 5.75 | 87.49 ± 0.73 | 55.43 ± 1.39 | 83.78 ± 2.55 | 75.56 ± 2.07 |
| $\beta = 0.01$ | 79.61 ± 0.60 | 75.59 ± 1.75 | 78.15 ± 0.07 | 74.84 ± 0.21 | 64.63 ± 0.38 | 87.74 ± 0.20 | 58.43 ± 0.41 | 85.14 ± 0.89 | 75.52 ± 0.56 |
| $\beta = 0.001$ | 79.52 ± 0.42 | 74.81 ± 0.97 | 78.20 ± 0.12 | 74.83 ± 0.26 | 64.85 ± 0.337 | 87.67 ± 0.14 | 58.02 ± 0.66 | 85.40 ± 0.81 | 75.41 ± 0.47 |

Table 22: AUPRC scores under different values of $k$ for truncated attention (in %).

| $k$ | Cora | Flickr | ACM | BlogCatalog | Facebook | Weibo | Reddit | Amazon | Average |
|---|---|---|---|---|---|---|---|---|---|
| 5 | 43.89 ± 1.53 | 37.52 ± 0.57 | 39.73 ± 0.11 | 34.75 ± 0.33 | 5.73 ± 1.15 | 59.73 ± 2.56 | 4.07 ± 0.18 | 61.09 ± 3.53 | 35.81 ± 1.25 |
| 10 | 44.18 ± 0.82 | 37.74 ± 0.38 | 39.73 ± 0.09 | 34.97 ± 0.34 | 5.67 ± 1.17 | 59.31 ± 2.80 | 4.05 ± 0.07 | 61.26 ± 3.47 | 35.86 ± 1.14 |
| $0.05n_{sup}$ | 43.94 ± 0.67 | 37.62 ± 0.29 | 39.74 ± 0.14 | 35.00 ± 0.32 | 5.61 ± 1.16 | 59.52 ± 2.46 | 4.06 ± 0.08 | 61.96 ± 3.23 | 35.93 ± 1.04 |
| $0.1n_{sup}$ | 44.08 ± 0.73 | 37.66 ± 0.30 | 39.76 ± 0.10 | 34.93 ± 0.28 | 5.53 ± 1.22 | 60.63 ± 2.49 | 4.06 ± 0.09 | 62.04 ± 2.34 | 36.09 ± 0.94 |
| $0.2n_{sup}$ | 44.04 ± 1.05 | 37.62 ± 0.34 | 39.73 ± 0.14 | 34.91 ± 0.59 | 5.16 ± 1.20 | 60.97 ± 0.85 | 4.06 ± 0.10 | 62.20 ± 3.12 | 36.09 ± 0.93 |
| $0.3n_{sup}$ | 43.55 ± 0.90 | 38.23 ± 0.26 | 39.14 ± 0.11 | 35.05 ± 0.37 | 5.87 ± 1.19 | 60.28 ± 0.68 | 3.92 ± 0.08 | 63.26 ± 2.48 | 36.16 ± 0.76 |
| $0.5n_{sup}$ | 43.93 ± 0.47 | 37.69 ± 0.26 | 39.74 ± 0.12 | 34.99 ± 0.31 | 5.61 ± 1.17 | 60.94 ± 0.21 | 4.05 ± 0.09 | 61.91 ± 3.05 | 36.11 ± 0.71 |

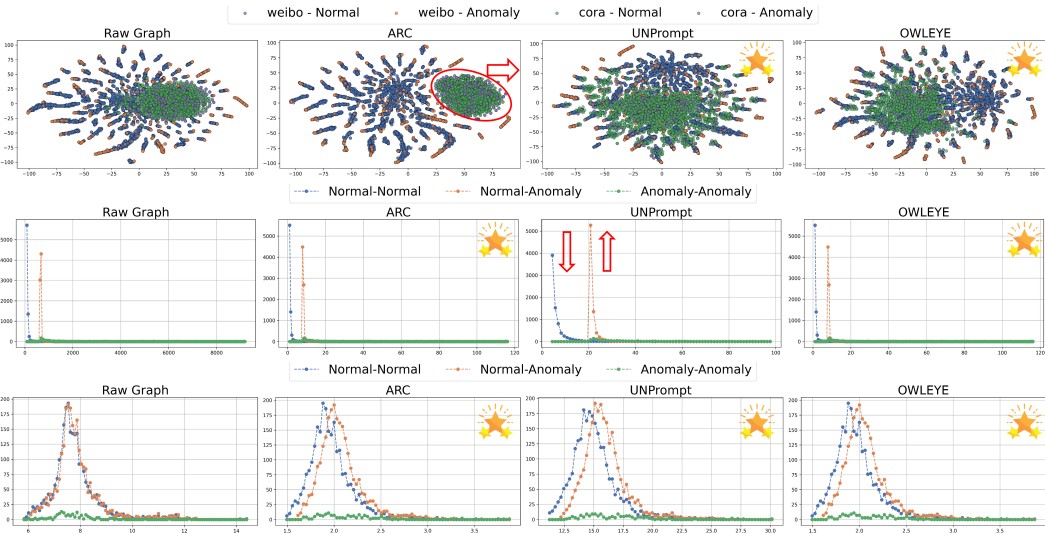

Figure 5: **Performance Visualization of SOTA GAD methods**, 🌠 denotes best. *Top row:* TSNE embeddings of Facebook and Weibo graph data for (a) original graph, (b) ARC, (c) UNPrompt, and (d) OWLEYE (ours). ARC pushes the two graphs apart rather than aligning them. *Middle and bottom rows:* pairwise Euclidean distances for Normal-Normal, Normal-Anomaly, and Anomaly-Anomaly node pairs on Weibo and Cora dataset, respectively. In the original graph (middle row, (a)), Normal-Normal pairs are denser than Normal-Anomaly pairs on Weibo dataset—an important pattern reversed by UNPrompt (middle row, (c)). The existing data preprocessing methods fail to either align the graphs into the share space or preserve important patterns after normalization.

## C.2 GRAPH SIMILARITY MEASUREMENT

We visualize the heat map for graph similarity measurement (defined in Equation 10) in Figure 6 (Left). Specifically, we measure the graph similarity between a test graph (e.g., Amazon, Reddit, Weibo, BlogCatalog, Facebook, ACM, Flickr, Cora) and a training graph (e.g., Pubmed, CiteSeer, Questions, YelpChi), where target_graph denotes the test graph where we extract and store patterns in $\text{Dict}_R^j$. For instance, for Amazon graph, we measure the similarity between Amazon with four graphs in the training set $\mathcal{T}_{train}$ and the patterns sampled from Amazon graph, denoted as target_graph. We observe that all graphs from the test sets heavily rely on the patterns extracted from its own graph, while some graphs, such as Weibo, may also leverages the information from other graphs (e.g., questions) to detect the anomalies.

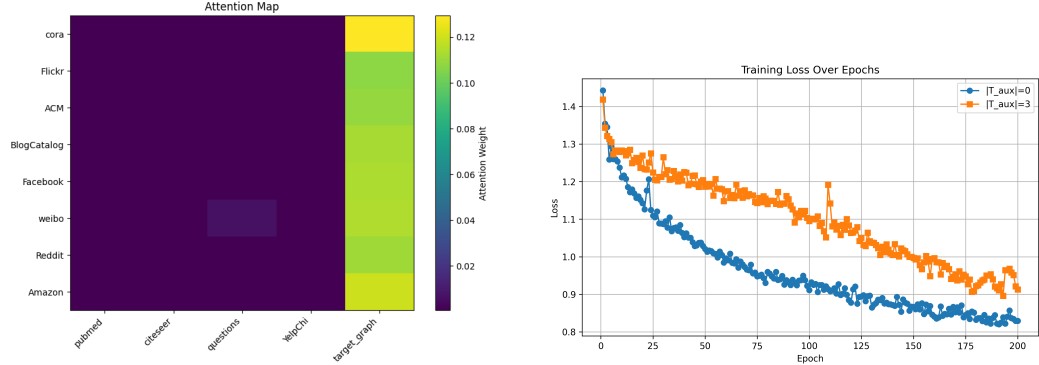

Figure 6: **Left:** Visualization of Graph Similarity Measurement for Different Graphs. **Right:** Training Loss vs Epochs

## C.3 TRAINING LOSS VS EPOCHS

We visualize the training loss vs the number of epochs in Figure 6 (Right) as the auxiliary information for case study 2. In this figure, we observe that when no graph from $\mathcal{T}_{aux}$ is used to finetune the model, the training loss (i.e., blue line) drops dramatically. However, when we add all three graphs from $\mathcal{T}_{aux}$ to finetune the model (i.e., the orange line), the training loss is much higher than that of model without using any graphs for model finetuning. This verify our conjecture that the reason that finetuning the model fail to achieve better performance is that the model is hard to train on too many graphs and thus hard to converge.

## C.4 ADDITIONAL ATTENTION MAP VISUALIZATION ON AMAZON DATASET

In this subsection, we provide a detailed visualization of the label matrices and cross-attention maps for the Amazon dataset to better understand how our model distinguishes normal nodes from anomalous ones. Similar to Figure 4, subfigures (a) and (b) display the cross-attention scores across the graphs, where the y-axis corresponds to the graph indices (0–3 indicating the four training graphs and 4 representing the test graph) and the x-axis denotes the ten extracted patterns learned for each graph. The top row shows the attribute attention and structural attention for a normal node and the bottom row shows the attribute attention and structural attention for an anomalous one. Subfigure (a) shows the attention map when the model makes the correct prediction, while Subfigure (b) presents the attention map when the model makes the incorrect prediction. Subfigure (c) in both figures presents the ground-truth label matrices that specify whether each pattern is normal or abnormal. Across both datasets, lighter colors such as light green and light yellow consistently indicate high similarity to the patterns associated with normal nodes, as reflected in the label matrices in Subfigure (c). The attention maps show a consistent pattern: correctly classified normal nodes have light-colored maps (high similarity to normal patterns), while correctly classified anomalies have darker maps (low similarity), and this pattern reverses for misclassified nodes.

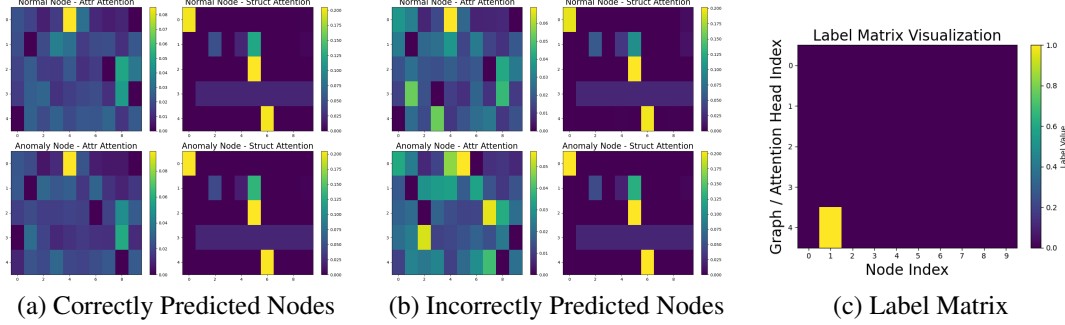

(a) Correctly Predicted Nodes     (b) Incorrectly Predicted Nodes     (c) Label Matrix

Figure 7: Visualization of cross-attention map on the Amazon dataset. Subfigures (a) and (b) display the cross-attention scores across the graphs, where the y-axis corresponds to the graph indices (0–3 indicating the four training graphs and 4 representing the test graph) and the x-axis denotes the ten extracted patterns learned for each graph. The top row shows the attribute attention and structural attention for a normal node and the bottom row shows the attribute attention and structural attention for an anomalous one. Subfigure (c) in both figures presents the ground-truth label matrices that specify whether each node is normal or abnormal.

## D    FUTURE DIRECTIONS

Learning on graphs has been a hot research topic in the machine learning community, evolving from early pattern-based and statistical graph mining methods to modern graph neural network models that enable end-to-end representation learning on complex relational data (Zhou et al., 2019; 2020a; 2022; Fu et al., 2022b; 2023; Zheng et al., 2024c;e; Fu et al., 2024; Li et al., 2025c; Wang et al., 2025). Despite substantial progress, emerging real-world applications increasingly demand graph learning frameworks that can handle heterogeneous and multi-modal signals, weak or noisy supervision, and rapidly evolving structures (Zheng et al., 2019; Zhou et al., 2020b; Zheng et al., 2021a; 2022; 2023; Ban et al., 2024; Tieu et al., 2024; Zheng et al., 2025b; Li et al., 2025a; Ning et al., 2025; Li et al., 2025b; Fu et al., 2025; Tieu et al., 2025a;b; He et al., 2025). In particular, multi-modal graph-based anomaly detection has become a critical direction, where relational patterns must be jointly modeled with node- and edge-level attributes from diverse sources such as text, time series, and images (Wu et al., 2024; Zheng et al., 2024a;b; Lin et al., 2024; Zou et al., 2025). Looking ahead, an important frontier lies in developing graph foundation models for anomaly detection, which aim to learn transferable, general-purpose representations from large-scale graphs and adapt them to downstream anomaly detection tasks under limited supervision (Zheng et al., 2024d; Lan et al., 2024). Such foundation models have the potential to unify graph structure, multi-modal semantics, and anomaly-specific inductive biases, enabling robust detection of rare and evolving anomalous behaviors across domains.

## E    BROADER IMPACT

This paper does not have significant potential negative societal impacts and it proposes a generalist anomaly detection model, which aims to detect anomalies in the unseen graphs. Our method can be applied to many applications, enhancing public safety by detecting fraud, cyberattacks, or suspicious activities.

