# OpenReview forum: "OwlEye: Zero-Shot Learner for Cross-Domain Graph Data Anomaly Detection"
_ICLR.cc/2026/Conference — ICLR 2026 Poster_

### Official Review · Reviewer_fjeF · 2025-10-25

**Soundness:** 3
**Presentation:** 2
**Contribution:** 2
**Rating:** 2
**Confidence:** 4

**Summary:**

The paper proposes OWLEYE, a framework for zero-shot graph anomaly detection across multiple domains. OWLEYE consists of three key modules: (1) cross-domain feature alignment to harmonize representations from graphs with heterogeneous features, (2) a multi-domain dictionary learning scheme to store and transfer both attribute and structure-based graph patterns, and (3) a truncated attention-based reconstruction module for robust anomaly inference in unlabeled, unseen graphs. Experiments demonstrate consistent performance gains over recent baselines, including state-of-the-art generalist and few-shot GAD methods, across a suite of diverse real-world graph benchmarks. The paper further investigates continual pattern augmentation and efficiency, positioning OWLEYE as a scalable, label-efficient, and extensible approach for cross-domain GAD.

**Strengths:**

**S1** The paper directly tackles the problem of cross-domain, zero-shot graph anomaly detection (GAD), a critical and timely challenge as graph datasets become more diverse and distributed in real applications.

**S2** The proposed pipeline—starting with feature projection/normalization (Section 2.1), followed by hierarchical pattern learning (Section 2.2), and concluding with a detailed in-context truncated attention mechanism (Section 2.3)—is thoughtfully constructed and mathematically grounded. The technical exposition of each module is presented with clarity and includes relevant equations (e.g., Equation for cross-domain normalization, attention, loss terms), supporting reproducibility.

**S3** The inclusion of comprehensive experimental results (see Table 1, Table 2, Table 6, Table 7), as well as meaningful visualizations (notably Figure 1), clearly demonstrate the effectiveness of OWLEYE in aligning cross-domain graphs and preserving crucial anomaly-detection patterns, something recent models like ARC and UNPrompt struggle with.

**Weaknesses:**

**W1**While the method is described mathematically, there is little depth to the theoretical guarantees for convergence or for the effectiveness of the normalization and attention truncation mechanisms. For example, the normalization equations on Page 4 define a cross-domain scaling scheme, but the paper does not provide a formal argument or sufficient empirical diagnostic to show that this process consistently preserves key structural semantics across arbitrary domains, beyond the handful of visual examples (Figure 1). Similarly, the attention-truncation approach is intuitively appealing but lacks any statistical justification concerning which anomalies could be missed or mistakenly preserved when top-k truncation is used, particularly as graph scale increases or distributions shift.

**W2**The core pattern storage and reconstruction process relies on “random sampling” nodes and computing truncated attention across dictionaries, but the sensitivity and robustness to this random selection are not systematically analyzed. Figure 4 visualizes attention distributions, but there is no supporting analysis as to how much predictive performance can vary across different random seeds or pattern samples, especially given the use of random support set extraction for the dictionary.

**W3**The method is described as storing “interpretable patterns” in dictionaries, but no concrete analysis or visualization is provided showing what these patterns are, how they differ across domains, or whether they correspond to genuine “normal” behaviors as claimed. Figure 2 shows the architecture, and Figure 4 provides a heat map, but neither supplies users with diagnostics to interpret failures or understand the significance of stored patterns, this is especially important in high-stakes GAD applications.

**W4**Although a broad set of baselines are used, the paper does not explain why several recently proposed cross-domain learning GNNs (many from the missing related work below) were not attempted as baselines, or even as ablation trims to OWLEYE.
Hyperparameter Sensitivity: While Table 10 (Appendix) selects some hyperparameter sweeps, there is little qualitative discussion of failure modes or special behavior, e.g., what happens if the size of the pattern dictionary is much smaller, or how performance degrades with heavy domain shift or label imbalance between training and test graphs.

**W5**Some mathematical definitions, e.g., in the attribute-level and structure-level pattern extraction (Section 2.2 equations), could use more explicit notation (e.g., variable range explanations) for clarity, especially for readers less familiar with GCN-based operations.

**W6**The cross-domain normalization claims median-based scaling (Equation Page 4) avoids domination by single datasets, but this is not shown in controlled experiments (e.g., single domain greatly divergent in distribution), how robust is this alignment when a domain is highly atypical? No “stress tests” or outlier cases are given.

**Questions:**

see weakness

---

> ### Author Response · Authors · 2025-11-14
> **Clarification for the Missing Part of the Review**
>
> Dear Reviewer fjeF,
>
> Thanks very much for your review! We are now carefully preparing the answer.
>
> For your W4, you mention "_Although a broad set of baselines are used, the paper does not explain why several recently proposed cross-domain learning GNNs (many from the missing related work below) were not attempted as baselines_", but the related work seems not given below.
>
> We are more than willing to analyze. Can you please list the related work you are referring to?
>
> Best,
>
> #12517 Authors

---

> ### Author Response · Authors · 2025-11-25
> **Rebuttal Answer from Authors (Part I)**
>
> Dear Reviewer fjeF,
>
> Thanks for your review! Your acknowledgment of our timely and critical motivation, solid and complete pipeline design, and comprehensive experiment results is truly appreciated.
>
> For each of your raised concerns, we prepared further detailed clarifications and additional extensive experiments.
>
> We updated the paper for your requested new experiments and analysis within the orange background color.
>
> We are more than happy to communicate with you to address any of your remaining concerns!
>
> > W1. While the method is described mathematically, there is little depth to the theoretical guarantees for convergence or for the effectiveness of the normalization and attention truncation mechanisms. For example, the normalization equations on Page 4 define a cross-domain scaling scheme, but the paper does not provide a formal argument or sufficient empirical diagnostic to show that this process consistently preserves key structural semantics across arbitrary domains, beyond the handful of visual examples (Figure 1). Similarly, the attention-truncation approach is intuitively appealing but lacks any statistical justification concerning which anomalies could be missed or mistakenly preserved when top-k truncation is used, particularly as graph scale increases or distributions shift.
>
> Thanks for your consideration. Regarding the two points mentioned in this weakness, i.e., _cross-domain feature normalization_ and _truncated attention_, we first give the general statement and then extend them from empirical and theoretical aspects to address your concerns deeply.
>
> #### [General Statement]
>
> In brief, (1) our feature alignment preserves key structural semantics across domains by maintaining the pairwise distance gaps between pairwise node pairs, ensuring cross-domain scaling does not distort critical patterns, which is a limitation observed in existing methods ARC [1] and UNPrompt [2]. (2) For the truncated attention, we provide a statistical justification: analogous to a trimmed mean estimator, discarding the top-k high-distance nodes bounds the influence of anomalies, ensuring that as long as $k$ exceeds the expected number of anomalies, the reconstructed features remain close to the true normal representation. Below are the more detailed explanations.
>
> #### [Detailed Explanation of Cross-Domain Feature Normalization]
> In Figure 1, we observe that a large distance gap between a Normal--Normal pair and a Normal--Anomaly pair on the Weibo dataset can be a crucial pattern for a GAD model to find a proper decision boundary to separate normal and anomalous nodes. This observation inspires us to leverage the pairwise node distance for Normal--Normal pairs, Normal--Anomaly pairs and Anomaly--Anomaly pairs. The existing feature preprocessing methods used in ARC and UNPrompt fail to leverage this kind of information and sometimes these feature preprocessing methods might distort the raw feature space. In contrast, our feature method could preserve this information. The experimental results in Table 11 indicate the superiority of our feature alignment method than ARC.
>
> In addition, we prepared the new visualization of the different feature preprocessing methods on Cora and Weibo datasets and added it in Appendix A.9.
> - Notice that Weibo has the largest feature norm (77.82), while the norms of other graphs are less than 3. To better visualize the different data distribution in the raw graphs across different domains, Weibo is selected.
> - We also selected Cora, as it is a citation network, different from Weibo as a social network.
> Similar to Figure 1, we have the similar observation in Figure 6 of Appendix A.9,
> - where ARC [1] pushes the two graphs apart rather than aligning them in the top row and UNPrompt [2] reverses the different patterns of Normal--Normal pairs pair and Normal--Anomaly pairs in the middle row;
> - In contrast, our method can fairly preserve the key graph structure even if two graphs are from different domains.

---

> ### Author Response · Authors · 2025-11-25
> **Rebuttal Answer from Authors (Part II)**
>
> #### [Detailed Explanation of Truncated Attention]
> For the experiments about the truncated attention method, we added additional experiments to show how the change of k-value impacts the performance. We have added the following results and analysis in Appendix A14.
>
> In the experiment, we set the value of k to p% of patterns in the dictionary where $n_{sup}$=1000 for each graph. We vary the percentage of it from 5% to 50% as well as two specific numbers of k (k=5 and k=10) and report the overall results across eight graphs below.
>
> The way to determine the value of k is based on the observation that the percentage of the anomaly is usually lower than 10%. Using the ratio of dictionary size can filter out most of the anomalies and then reduce the uncertainty.
>
> In a situation where the percentage of anomalies is unusually high, we can manually increase the value of k. The results show that when we set the value of k to 10% to 30% of patterns in the dictionary, the AUPRC scores are around 36.1% and its performance decreases if we increase the percentage to 50%. One possible explanation is that we filter out too many patterns that might be useful for identifying anomalies.
>
> Similarly, when we decreased the value of k to a value that could not filter out most of the anomalies, the performance starts to decrease, as the method involves the anomalies for feature construction.
>
> | Dataset     | k=5               | k=10              | k=0.05*n_{sup}         | k=0.1*n_{sup}         | k=0.2*n_{sup}         | k=0.3*n_{sup}          | k=0.5*n_{sup}         |
> |-------------|-------------------|-------------------|-----------------------|-----------------------|-----------------------|-----------------------|-----------------------|
> | Cora        | 0.4389 ± 0.0153   | 0.4418 ± 0.0082   | 0.4394 ± 0.0067       | 0.4408 ± 0.0073       | 0.4404 ± 0.0105       | 0.4355 ± 0.0090       | 0.4393 ± 0.0047       |
> | Flickr      | 0.3752 ± 0.0057   | 0.3774 ± 0.0038   | 0.3762 ± 0.0029       | 0.3766 ± 0.0030       | 0.3762 ± 0.0034       | 0.3823 ± 0.0026       | 0.3769 ± 0.0026       |
> | ACM         | 0.3973 ± 0.0011   | 0.3973 ± 0.0009   | 0.3974 ± 0.0014       | 0.3976 ± 0.0010       | 0.3973 ± 0.0014       | 0.3914 ± 0.0011       | 0.3974 ± 0.0012       |
> | BlogCatalog | 0.3475 ± 0.0033   | 0.3497 ± 0.0034   | 0.3500 ± 0.0032       | 0.3493 ± 0.0028       | 0.3491 ± 0.0059       | 0.3505 ± 0.0037       | 0.3499 ± 0.0031       |
> | Facebook    | 0.0573 ± 0.0115   | 0.0567 ± 0.0117   | 0.0561 ± 0.0116       | 0.0553 ± 0.0122       | 0.0516 ± 0.0120       | 0.0587 ± 0.0119       | 0.0561 ± 0.0117       |
> | Weibo       | 0.5973 ± 0.0256   | 0.5931 ± 0.0280   | 0.5952 ± 0.0246       | 0.6063 ± 0.0249       | 0.6097 ± 0.0085       | 0.6028 ± 0.0068       | 0.6094 ± 0.0021       |
> | Reddit      | 0.0407 ± 0.0018   | 0.0405 ± 0.0007   | 0.0406 ± 0.0008       | 0.0406 ± 0.0009       | 0.0406 ± 0.0010       | 0.0392 ± 0.0008       | 0.0405 ± 0.0009       |
> | Amazon      | 0.6109 ± 0.0353   | 0.6126 ± 0.0347   | 0.6196 ± 0.0323       | 0.6204 ± 0.0234       | 0.6220 ± 0.0312       | 0.6326 ± 0.0248       | 0.6191 ± 0.0305       |
> | **Average** | **0.3581 ± 0.0125** | **0.3586 ± 0.0114** | **0.3593 ± 0.0104**   | **0.3609 ± 0.0094**   | **0.3609 ± 0.0093**   | **0.3616 ± 0.0076**   | **0.3611 ± 0.0071**   |
>
> As for the theoretical analysis, here is the explanation of why the truncated attention mechanism works.
>
> Following [3], for a set of observations with a fraction $\epsilon$ of outliers, the _**trimmed mean estimator**_ $\hat{\mu}$​ satisfies:
>
> $|\hat{\mu}\_\text{trim} - \mu\_\text{true}| \leq O(\epsilon \sigma) $
>
> where $\sigma^2$ is the variance of the normal distribution $F\_\text{normal}$​.
>
> This illustrates that removing the largest deviations before aggregation yields an estimator that remains close to the true normal mean even under adversarial contamination.
>
> In analogy, truncated attention behaves as a neural trimmed estimator: by discarding high-distance (thus likely abnormal) nodes before reconstruction, it bounds the influence of outliers on the learned representation.
>
> Hence, it implies that as long as the number of truncated nodes $k$ exceeds the expected number of anomalies ($\epsilon |V|$), the reconstructed feature representation is statistically close to the true mean normal representation.
>
> Therefore, it provides a robustness guarantee against mislabeled pseudo-support nodes.
>
> Reference:
>
> [1] Liu et al.,  ARC: A Generalist Graph Anomaly Detector with In-Context Learning. NeurIPS 2024
>
> [2] Niu et al., Zero-shot Generalist Graph Anomaly Detection with Unified Neighborhood Prompts. IJCAI 2025
>
> [3] Huber, Peter J. "Robust estimation of a location parameter." In Breakthroughs in statistics: Methodology and distribution, Springer New York, 1992.

---

> > ### Author Response · Authors · 2025-11-25
> > **Rebuttal Answer from Authors (Part III)**
> >
> > > W2. The core pattern storage and reconstruction process relies on “random sampling” nodes and computing truncated attention across dictionaries, but the sensitivity and robustness to this random selection are not systematically analyzed. Figure 4 visualizes attention distributions, but there is no supporting analysis as to how much predictive performance can vary across different random seeds or pattern samples, especially given the use of random support set extraction for the dictionary.
> >
> > Thanks for your consideration, but please allow us to clarify that we have sensitivity and robustness analysis for the random sampling. To be specific:
> >
> > - For the methodological design, in line 234, we mention that we randomly extract $n_{sup}$ patterns from each graph and store them in two dictionaries.
> >
> > - In the corresponding experiment, we report the average AUROC/AUPRC with standard deviations across 5 trials with different random seeds for the sampling. The standard deviation in Table 1 and Table 2 shows the randomness of "random sampling" and quantifies how much predictive performance can vary across different random seeds.
> >
> > ---
> >
> > > W3. The method is described as storing “interpretable patterns” in dictionaries, but no concrete analysis or visualization is provided showing what these patterns are, how they differ across domains, or whether they correspond to genuine “normal” behaviors as claimed. Figure 2 shows the architecture, and Figure 4 provides a heat map, but neither supplies users with diagnostics to interpret failures or understand the significance of stored patterns, this is especially important in high-stakes GAD applications.
> >
> > First of all, please allow us to clarify that we do not claim “interpretable patterns” stored in dictionaries in our paper.
> >
> > Second, in the conclusion, we claim that the proposed OWLEYE is an interpretable solution. The reason behind this is our proposed truncated attention mechanism.
> >
> > To show that, we prepare additional visualizations in Appendix A.10 (Figure 7 and 8) to demonstrate that the interpretability of our method via the attention map by visualizing the correct prediction on normal node and anomalous node and the incorrect prediction on normal node and anomalous node on the Cora and Amazon datasets.
> > - In the visualization of attention maps, lighter colors such as light green and light yellow consistently indicate high similarity to the patterns associated with normal nodes, as reflected in the label matrices in Figure 7 and 8.
> > - By examining these visualizations, we observe a clear and consistent relationship between the attention intensity and the correctness of the model’s predictions:
> >   - when the model correctly identifies a normal node, its attention map is dominated by light colors, suggesting a strong similarity to normal patterns stored in the dictionary;
> >   - when it correctly identifies an anomalous node, the attention map becomes noticeably darker, indicating low similarity to normal behavior.
> > - Importantly, this trend reverses for misclassified nodes: normal nodes that are wrongly predicted as anomalies exhibit darker color in attention map, while misclassified anomalous nodes show lighter colors, showing the high similarity to those of normal nodes.
> > - This systematic behavior demonstrates that the attention map offers an intuitive and faithful interpretation mechanism, as the color patterns directly reflect whether the node under consideration resembles the learned normal patterns, thereby revealing both the reasoning behind correct predictions and the failure modes behind incorrect ones.

---

> > > ### Author Response · Authors · 2025-11-25
> > > **Rebuttal Answer from Authors (Part IV)**
> > >
> > > > W4. Although a broad set of baselines are used, the paper does not explain why several recently proposed cross-domain learning GNNs (many from the missing related work below) were not attempted as baselines, or even as ablation trims to OWLEYE. Hyperparameter Sensitivity: While Table 10 (Appendix) selects some hyperparameter sweeps, there is little qualitative discussion of failure modes or special behavior, e.g., what happens if the size of the pattern dictionary is much smaller, or how performance degrades with heavy domain shift or label imbalance between training and test graphs.
> > >
> > > First of all, we have added more related work discussion in Section 4 and Appendix B. Moreover, we are more than willing to discuss and analyze your recommended related work. The review seems incomplete for this part, please let us know your suggestions and we will add more as soon as possible.
> > >
> > > Next, we answer three additional experiments you are interested in.
> > >
> > > #### [Experiments of Small Dictionary Size]
> > >
> > > In Table 5, we demonstrate that when the size of patterns extracted from each graph is reduced from 2000 to 10 as shown below, the performance (AUPRC) of our method decreases by less than 1\%  on average. This shows that even if the size of the dictionary is small, the performance of our method is robust.
> > >
> > > | n_sup | Cora | Flickr | ACM | BlogCatalog | Facebook | Weibo | Reddit | Amazon | Average |
> > > |------|----------------|--------|-----|-------------|----------|--------|--------|--------|----------|
> > > | 10   | 44.55 ± 0.48 | 37.86 ± 0.26 | 39.57 ± 0.08 | 34.84 ± 0.33 | 5.63 ± 0.73 | 61.30 ± 0.44 | 4.03 ± 0.12 | 55.89 ± 2.16 | 35.46 ± 0.58 |
> > > | 100  | 43.52 ± 0.78 | 37.45 ± 0.23 | 39.87 ± 0.06 | 34.76 ± 0.38 | 5.43 ± 1.34 | 61.13 ± 0.67 | 4.03 ± 0.08 | 60.49 ± 2.82 | 35.84 ± 0.79 |
> > > | 200  | 43.24 ± 0.50 | 37.65 ± 0.29 | 39.92 ± 0.12 | 34.72 ± 0.35 | 5.32 ± 1.40 | 60.70 ± 0.53 | 4.02 ± 0.08 | 62.51 ± 1.45 | 36.01 ± 0.59 |
> > > | 500  | 43.28 ± 0.92 | 37.96 ± 0.39 | 39.86 ± 0.13 | 34.79 ± 0.35 | 5.43 ± 1.32 | 61.00 ± 0.47 | 4.04 ± 0.11 | 62.77 ± 1.36 | 36.14 ± 0.63 |
> > > | 1000 | 43.88 ± 0.73 | 37.99 ± 0.33 | 39.82 ± 0.08 | 34.92 ± 0.39 | 5.33 ± 1.29 | 60.52 ± 0.79 | 4.03 ± 0.12 | 61.76 ± 2.50 | 36.03 ± 0.78 |
> > > | 2000 | 43.94 ± 0.46 | 37.69 ± 0.25 | 39.75 ± 0.13 | 34.99 ± 0.31 | 5.62 ± 1.17 | 60.90 ± 0.21 | 4.25 ± 0.11 | 62.20 ± 3.18 | **36.17 ± 0.73** |
> > >
> > > #### [Experiments of Heavy Domain Shift]
> > >
> > > Further, we evaluate our method under a heavy domain shift setting. We have added the following results in Appendix A.6.
> > >
> > > To simulate a scenario where the e-commerce co-review domain is entirely unseen during training, we remove the YelpChi graph from the training set. This creates a substantial distribution mismatch, as no graph from the e-commerce co-review domain is available during model learning.
> > >
> > > The experimental results reveal that, under this heavy domain shift, the performance on the Amazon graph (also from the e-commerce co-review domain) decreases by 4.6\% in AUPRC, while its AUROC remains largely unchanged.
> > >
> > > This behavior is expected: with an entire domain missing during training, domain-specific patterns become harder to recover. Nevertheless, the performance drop remains moderate, demonstrating that the heterogeneous patterns learned from the remaining graphs still generalize well enough to support robust anomaly detection on the Amazon dataset.
> > >
> > > Overall, these findings indicate that while heavy domain shift does lead to reasonable performance degradation for the affected domain, the cross-domain structural and attribute patterns captured by our model continue to provide meaningful generalization.

---

> > > > ### Author Response · Authors · 2025-11-25
> > > > **Rebuttal Answer from Authors (Part V)**
> > > >
> > > > | Dataset     | Pubmed, CiteSeer, Questions, YelpChi (AUROC) | Pubmed, CiteSeer, Questions, YelpChi (AUPRC) | Pubmed, CiteSeer, Questions (AUROC) | Pubmed, CiteSeer, Questions (AUPRC) |
> > > > |-------------|----------------------------------------------|-----------------------------------------------|--------------------------------------|--------------------------------------|
> > > > |             | mean ± std                                   | mean ± std                                    | mean ± std                           | mean ± std                           |
> > > > | Cora        | 0.7952 ± 0.0043                              | 0.4394 ± 0.0046                               | 0.7989 ± 0.0027                      | 0.4476 ± 0.0051                      |
> > > > | Flickr      | 0.7481 ± 0.0097                              | 0.3769 ± 0.0025                               | 0.7636 ± 0.0255                      | 0.3791 ± 0.0038                      |
> > > > | ACM         | 0.7820 ± 0.0012                              | 0.3975 ± 0.0013                               | 0.7893 ± 0.0087                      | 0.3972 ± 0.0015                      |
> > > > | BlogCatalog | 0.7483 ± 0.0026                              | 0.3499 ± 0.0031                               | 0.7563 ± 0.0146                      | 0.3504 ± 0.0036                      |
> > > > | Facebook    | 0.6485 ± 0.0037                              | 0.0562 ± 0.0117                               | 0.6627 ± 0.0097                      | 0.0619 ± 0.0137                      |
> > > > | Weibo       | 0.8765 ± 0.0014                              | 0.6090 ± 0.0021                               | 0.8810 ± 0.0022                      | 0.6118 ± 0.0104                      |
> > > > | Reddit      | 0.5802 ± 0.0068                              | 0.0425 ± 0.0011                               | 0.5779 ± 0.0022                      | 0.0397 ± 0.0006                      |
> > > > | Amazon      | 0.8543 ± 0.0079                              | 0.6220 ± 0.0318                               | 0.8515 ± 0.0080                      | 0.5764 ± 0.0375                      |
> > > >
> > > > #### [Experiments of Label Imbalance between Training and Test Graphs]
> > > >
> > > > As for the label imbalance experiment, 12 datasets used in our experiment have different anomaly rates, ranging from 2.31% to 10.3%.
> > > >
> > > > For the graphs in the training set, the anomaly rates are 3.04% (PubMed), 4.50% (CiteSeer), 2.98% (Questions), and 5.10% (YelpChi).
> > > >
> > > > For the graphs in the test set, the anomaly rates are 6.76% (Amazon), 3.33% (Reddit), 10.3% (Weibo), 2.31% (Facebook), 5.94% (Flickr), 5.77% (BlogCatalog), 3.62% (ACM), and 5.53%(Cora).
> > > >
> > > > In addition, Table 12 in Appendix A.5 shows that when our method consistently outperforms two baseline methods when the anomalies rate increases to 23.31%.
> > > >
> > > > ---
> > > >
> > > > > W5. Some mathematical definitions, e.g., in the attribute-level and structure-level pattern extraction (Section 2.2 equations), could use more explicit notation (e.g., variable range explanations) for clarity, especially for readers less familiar with GCN-based operations.
> > > >
> > > > Thanks for the consideration. We have updated the entire Section 2.2 by adding more context and notations, and highlighting them by orange background color. The updated version is more friendly to GCN beginners.

---

> > > > > ### Author Response · Authors · 2025-11-25
> > > > > **Rebuttal Answer from Authors (Part VI)**
> > > > >
> > > > > > W6. The cross-domain normalization claims median-based scaling (Equation Page 4) avoids domination by single datasets, but this is not shown in controlled experiments (e.g., single domain greatly divergent in distribution), how robust is this alignment when a domain is highly atypical? No “stress tests” or outlier cases are given.
> > > > >
> > > > > As requested, we prepared the mean operation versus median operation during the feature processing stage, and the experiment of robustness when a domain is highly atypical. We added these two experiments in Appendix A.16 and A.17.
> > > > >
> > > > > First, we compute the norm of raw features of each graph for background knowledge. Notice that the norm of weibo is significantly larger than other graphs. We compare the median and Mean operation in our feature preprocessing module and report the results below.
> > > > >
> > > > >
> > > > > | Dataset    | Norm     |
> > > > > | -------- | --------|
> > > > > | Pubmed |   0.24 |
> > > > > | Questions |  0.45 |
> > > > > | Weibo  |  **77.82** |
> > > > > | YelpChi |   0.07 |
> > > > > | Cora |   2.41 |
> > > > > | Flickr |   0.25 |
> > > > > | ACM |   0.10 |
> > > > > | BlogCatalog |  0.30 |
> > > > > | Facebook |   2.38 |
> > > > > | Citeseer |   2.81 |
> > > > > | Reddit |   0.02 |
> > > > > | Amazon |   0.49 |
> > > > >
> > > > >
> > > > > After setting Pubmed, Questions, Weibo and YelpChi as the training graph and the rest graphs are used for testing, the results below show that replacing Median with Mean operation significantly reduces the performance on  Cora, CiteSeer and Amazon datasets, indicating that Mean operation results in one graph dominating the norm and leads to performance degradation.
> > > > >
> > > > > | Dataset      | Median (AUPRC %)        | Mean (AUPRC %)          |
> > > > > |--------------|--------------------------|---------------------------|
> > > > > | Cora         | 43.63 ± 1.94             | 34.95 ± 11.96             |
> > > > > | Flickr       | 37.92 ± 0.42             | 37.64 ± 1.15              |
> > > > > | ACM          | 39.55 ± 0.12             | 39.05 ± 0.31              |
> > > > > | BlogCatalog  | 35.00 ± 0.20             | 34.53 ± 0.63              |
> > > > > | Facebook     | 4.56 ± 0.71              | 4.30 ± 1.15               |
> > > > > | CiteSeer     | 42.57 ± 1.05             | 33.65 ± 12.64             |
> > > > > | Reddit       | 4.19 ± 0.14              | 3.98 ± 0.16               |
> > > > > | Amazon       | 56.07 ± 2.28             | 38.56 ± 17.31             |
> > > > > | **Average**  | 32.94 ± 0.86             | 28.33 ± 5.66              |
> > > > >
> > > > > Next, we aim to examine if our method is robust when a domain is highly atypical by varying the training graph.
> > > > >
> > > > > Since Weibo has the largest norm, we do the following two experiments with different training set:
> > > > > - 1) In the first experiment, we select Pubmed, Questions, Weibo and YelpChi as the training set, denoted as “With Weibo”;
> > > > > - 2) In the second experiment, we select Pubmed, Questions, CiteSeer and YelpChi as the training set, denoted as “With CiteSeer”.
> > > > >
> > > > > As shown in the table below, we observe that the performances of the most datasets are consistent in two settings.
> > > > >
> > > > > | Dataset      | With Weibo (AUPRC %)     | With CiteSeer (AUPRC %)  |
> > > > > |--------------|---------------------------|----------------------------|
> > > > > | Cora               | 43.63 ± 1.94              | 43.94 ± 0.46               |
> > > > > | Flickr              | 37.92 ± 0.42              | 37.69 ± 0.25               |
> > > > > | ACM              | 39.55 ± 0.12              | 39.75 ± 0.13               |
> > > > > | BlogCatalog   | 35.00 ± 0.20              | 34.99 ± 0.31               |
> > > > > | Facebook       | 4.56 ± 0.71               | 5.62 ± 1.17                |
> > > > > | Reddit            | 4.19 ± 0.14               | 4.25 ± 0.11                |
> > > > > | Amazon         | 56.07 ± 2.28              | 62.20 ± 3.18               |
> > > > > | **Average**   | 31.56 ± 0.83              | 32.63 ± 0.80               |

---

### Official Review · Reviewer_aCfo · 2025-10-31

**Soundness:** 2
**Presentation:** 3
**Contribution:** 2
**Rating:** 4
**Confidence:** 3

**Summary:**

This paper proposes a novel zero-shot graph anomaly detection framework called OWLEYE. The core idea of ​​OWLEYE is to learn and store representative patterns of normal behavior from multiple source graphs into a structured dictionary, thereby effectively detecting anomalies in unknown graph data without retraining.

**Strengths:**

1. This paper introduces a dynamic dictionary to store attribute-level and structure-level normal patterns, supporting incremental knowledge updates.

2. Avoiding noise pollution from spurious normal node sampling is a crucial issue. Truncated attention mechanisms, as a soft filtering method, are more robust than random sampling or hard thresholding.

3. Visualized feature analysis demonstrates a deep understanding of the data.

**Weaknesses:**

1. This paper claims that the runcated attention mechanism "filters out potential abnormal nodes," but it doesn't provide theoretical analysis or comparisons with other attention mechanisms.

2. Dictionary learning mechanisms implicitly assume "pattern transferability," and I have concerns about the general applicability of this assumption.

3. PCA is an unsupervised linear dimensionality reduction method. It can only guarantee dimensionality uniformity, but it cannot guarantee deep semantic alignment of features from different domains. This seems to imply the same assumption as the previous question: "normal patterns in different domains are similar."

**Questions:**

Please refer to the weaknesses.

---

> ### Author Response · Authors · 2025-11-25
> **Rebuttal Answer from Authors (Part I)**
>
> Dear Reviewer aCfo,
>
> Thanks very much for your review! We are very pleased to learn your positive feedback on our truncated attention design and empirical visualization analysis.
>
> For each of your raised weaknesses, we prepared the detailed answer below. Accordingly, we have updated the paper pdf file, and new content is marked by the orange background color.
>
> > W1. This paper claims that the truncated attention mechanism "filters out potential abnormal nodes," but it doesn't provide theoretical analysis or comparisons with other attention mechanisms.
>
> Thanks for the consideration. We will extend the statement from both empirical and theoretical aspects to address your concern.
>
> Empirically, in the ablation study of Section 3.2, we compare truncated attention with standard attention (self-attention of Transformer) mechanism (OWLEYE-T). The experiment shows that our proposed truncated attention can improve around 1.8\% over the self-attention mechanism.
>
> Theoretically, we prepared an explanation of why the truncated attention mechanism works.
>
> Following [1], for a set of observations with a fraction $\epsilon$ of outliers, the _**trimmed mean estimator**_ $\hat{\mu}$​ satisfies:
>
> $|\hat{\mu}\_\text{trim} - \mu\_\text{true}| \leq O(\epsilon \sigma) $
>
> where $\sigma^2$ is the variance of the normal distribution $F\_\text{normal}$​.
>
> This illustrates that removing the largest deviations before aggregation yields an estimator that remains close to the true normal mean even under adversarial contamination.
>
> In analogy, truncated attention behaves as a neural trimmed estimator: by discarding high-distance (thus likely abnormal) nodes before reconstruction, it bounds the influence of outliers on the learned representation.
>
> Hence, it implies that as long as the number of truncated nodes $k$ exceeds the expected number of anomalies ($\epsilon |V|$), the reconstructed feature representation is statistically close to the true mean normal representation.
>
> Therefore, it provides a robustness guarantee against mislabeled pseudo-support nodes.
>
> Reference:
>
> [1] Huber, Peter J. "Robust estimation of a location parameter." In Breakthroughs in statistics: Methodology and distribution, 1992.

---

> ### Author Response · Authors · 2025-11-25
> **Rebuttal Answer from Authors (Part II)**
>
> > W2. Dictionary learning mechanisms implicitly assume "pattern transferability," and I have concerns about the general applicability of this assumption.
>
> Thanks for the consideration.
>
> To begin with, the transferable patterns denote the common knowledge across domains, which is a common and foundational assumption for most domain adaptation methods [1].
>
> Next, we will empirically demonstrate its effectiveness from three aspects.
>
> **First**, we verify that the patterns extracted from one graph are valuable in identifying anomalies in another graph in the ablation study (Table 3 and Table 4), based on the real-world relational graph datasets.
> - Table 3 shows a consistent performance improvement as more patterns from other graphs are incorporated, validating OWLEYE’s ability to incrementally learn from new data sources in a plug-and-play fashion. Similarly, we can draw a similar conclusion from Table 4.
>
> **Second**, in Figure 4 in the appendix, we also visualize the attention map for graph similarity measurement, where target_graph denotes the test graph where we extract and store patterns in $Dict_j^R$.
> - We observe that all graphs from the test sets heavily rely on the patterns extracted from its own graph, while some graphs, such as Weibo, benefit from the patterns extracted from other graphs (e.g., questions) to detect the anomalies.
>
> **Third**, below are the additional results w.r.t. AUPRC to show whether the patterns from other graphs can benefit the anomalies detection. The results show that when patterns are extracted from multiple graphs, the performance is improved by 0.32% on average across eight graphs. We have added the following results in Appendix A.15.
>
> | Dataset        | Patterns from target graph only (%) | Patterns from multiple graphs (%) | Improvement (%) |
> |----------------|--------------------------------------|------------------------------------|------------------|
> | Cora        | 43.21 ± 0.98                         | 43.94 ± 0.46                       | 0.73             |
> | Flickr      | 37.41 ± 0.16                         | 37.69 ± 0.25                       | 0.28             |
> | ACM         | 39.84 ± 0.23                         | 39.75 ± 0.13                       | -0.09            |
> | BlogCatalog | 34.81 ± 0.44                         | 34.99 ± 0.31                       | 0.17             |
> | Facebook    | 5.27 ± 1.34                          | 5.62 ± 1.17                        | 0.35             |
> | Weibo       | 60.47 ± 0.75                         | 60.90 ± 0.21                       | 0.43             |
> | Reddit      | 4.03 ± 0.08                          | 4.25 ± 0.11                        | 0.23             |
> | Amazon      | 61.72 ± 3.01                         | 62.20 ± 3.18                       | 0.48             |
> | **Average**     | 35.84 ± 0.88                         | **36.17 ± 0.73**                       | 0.32             |
>
> Reference:
>
> [1] Ben-David, Shai, John Blitzer, Koby Crammer, Alex Kulesza, Fernando Pereira, and Jennifer Wortman Vaughan. "A theory of learning from different domains." Machine learning 79, no. 1 (2010): 151-175.

---

> ### Author Response · Authors · 2025-11-25
> **Rebuttal Answer from Authors (Part III)**
>
> > W3. PCA is an unsupervised linear dimensionality reduction method. It can only guarantee dimensionality uniformity, but it cannot guarantee deep semantic alignment of features from different domains. This seems to imply the same assumption as the previous question: "normal patterns in different domains are similar."
>
> First of all, please allow us to clarify that we do not claim that _``normal patterns in different domains are similar’’_.
>
> Next, we extend it as follows.
>
> In Section 2.1, we first propose a key research question: how can we effectively unify and align heterogeneous features from diverse graph domains without compromising their semantic integrity? In other words, we emphasize that heterogeneous features from different graph domains should be aligned without compromising their semantic integrity, which means we cannot assume that normal patterns across domains are inherently similar.
>
> Although PCA or other dimensionality-consistent preprocessing can enforce uniform feature size, the semantic meaning of each dimension remains distinct across datasets. As our Figure 1 and Figure 6 show:
> - the previous state-of-the-art baseline ARC [1] tends to separate graphs rather than align them,
> - and the newest state-of-the-art baseline UNPrompt [2] can distort the input space, reversing density relationships between Normal--Normal and Normal--Anomaly pairs.
>
> To address this, we propose cross-domain feature normalization in Section 2.2 that aligns distributions across graphs while preserving critical pairwise node distances, ensuring pairwise node semantics essential for anomaly detection are maintained.
> - We first validate this in the visualization in FIgure 1 and Figure 6, where the TSNE visualization of the graph after feature normalization by our method is similar to that of the raw graph.
> - Then, we further demonstrate its effectiveness in multiple experiments.
>   - For instance, in the ablation study (Figure 3), we evaluate the impact of removing feature normalization (OWLEYE vs OWLEYE-N), and the experimental results show more than 0.7\% performance degradation on average across 8 test graphs w.r.t AUPRC.
>   - The experiment in Appendix A.4 shows that our proposed method can improve ARC by 1\% on average after replacing its feature preprocessing with our feature alignment method.
>
> Reference:
>
> [1] Liu et al.,  ARC: A Generalist Graph Anomaly Detector with In-Context Learning. NeurIPS 2024
>
> [2] Niu et al., Zero-shot Generalist Graph Anomaly Detection with Unified Neighborhood Prompts. IJCAI 2025

---

> > ### Comment · Reviewer_aCfo · 2025-11-27
> >
> > Thanks for the replies. These discussions have made this paper much clearer. After reading the comments from other reviewers, I decided to raise my score.

---

### Official Review · Reviewer_fw57 · 2025-10-31

**Soundness:** 3
**Presentation:** 2
**Contribution:** 3
**Rating:** 6
**Confidence:** 4

**Summary:**

In this paper, the authors propose OWLEYE, an in-context graph anomaly detection model. OWLEYE’s design is based on three main components: (1) a PCA-based projection with normalization to align features across graphs; (2) a multi-domain dictionary learning approach that extracts attribute and structural patterns from graphs using GNNs; and (3) an anomaly score computation method based on reconstructing input graph patterns using the dictionary. The experiments compare OWLEYE against multiple supervised, unsupervised, and in-context (ARC and UNPrompt) graph anomaly detection methods using eight traditional benchmarks. The results show that OWLEYE achieves the best results on 3/8 datasets in the zero-shot setting and 4/8 datasets in the 10-shot setting. Additional results include an ablation study, an efficiency analysis, and case studies on pattern augmentation, fine-tuning, and dictionary size.

**Strengths:**

- The paper addresses a relevant and recent problem

- The idea of using dictionary learning for in-context GAD is interesting

- The proposed solution outperforms the baselines in most settings

**Weaknesses:**

- The paper writing could be greatly improved: Several parts of the paper concern me. Figure 1 is hard to read because of the small font and markers, and the overwhelming amount of information overall. The feature alignment method proposed in Section 2.1 is not well justified (why is it better than other alignment approaches?).  The loss from Equation 13 is also not well motivated (is it novel? Why is each element in the loss needed?). In general, Section 2 should include citations for all ideas that were inspired by previous work. Figure 3 can be misleading because some of the differences seem to be lower than 1%. The related work is very brief and doesn’t cover relevant literature, such as cross-domain graph anomaly detection, which can be unsupervised.

- The problem setting is not completely clear: It is not clear what information about the test graph is available. In NLP, in-context learning usually assumes that only examples are given (not the entire dataset). However, this paper seems to assume that the entire graph is available during testing. The availability of the test graph makes a big difference, even without labels, as shown in the research on unsupervised GAD. My intuition is that in this setting, it makes more sense to start from an unsupervised method and then try to improve it by incorporating additional information from other labeled datasets (the graph topology is likely more informative than the 10 examples given as in-context input). Notice that if the test graph is not fully observed, the problem becomes much harder, and the approach I described does not work.

- The experimental results need better insights: In the ablation study, it is not clear what the replacement is for each component of OWLEYE. For instance, ARC applies SVD and ordering instead of PCA and normalization for feature alignment. Anomaly score is computed using attention over in-context examples in ARC. The paper should make a case that each component of ARC is better than the alternative. It is also unclear why 200 patterns are sufficient to identify the anomalies; visualizing the patterns and anomalies could be helpful to understand what is happening (not clear if that is what is done in Fig. 1). There is no discussion on training time, which could be a deciding factor for practitioners.

**Questions:**

1) Why is the feature alignment method proposed in Section 2.1 better than existing approaches?

2) Is the loss from Equation 13 novel? What is the motivation?

3) What are the values (numbers) in Figure 3?

4) What information about the test graph is available in the problem setting considered?

5) What is the replacement used for each component of OWLEYE in the ablation study? Are these components the state of the art?

---

> ### Author Response · Authors · 2025-11-25
> **Rebuttal Answer from Authors (Part I)**
>
> Dear Reviewer fw57,
>
> Thanks very much for your review! We are glad to learn your appreciation of our dictionary learning and in-context learning idea for graph anomaly detection, and your acknowledgement of our effective performance.
>
> For your raised suggestions, we prepared additional explanations and experiment analysis, and we uploaded the updated paper accordingly. The revised part is marked with the orange background color.
>
> Since the questions you asked are mainly originated from the weaknesses you suggested, we prepared the detailed answer in the QA format and then answer remaining points in weaknesses.
>
> > Q1. Why is the feature alignment method proposed in Section 2.1 better than existing approaches?
>
> In short, our feature alignment method is better than the existing feature preprocessing in ARC [1] and UNPrompt [2] because it preserves the crucial semantic gap between pairwise node distances, which is often distorted by prior methods and is essential for establishing a reliable decision boundary.
>
> We extend it as follows.
>
> As the visualization shown in Figure 1, we observe that a large distance gap between a Normal--Normal pair and a Normal--Anomaly pair on the Weibo dataset can be a crucial pattern for a GAD model to find a proper decision boundary to separate normal and anomalous nodes.
>
> This observation inspires us to leverage the pairwise node distance for Normal-Normal pairs, Normal--Anomaly pairs and Anomaly--Anomaly pairs. The existing feature preprocessing methods used in ARC and UNPrompt fail to leverage this kind of information and sometimes these feature preprocessing methods might distort the raw feature space, as shown in Figure 1.
> Our proposed feature alignment method preserves this important semantic information. The experimental results in Table 11 indicate the superiority of our feature alignment method than ARC.
>
> Reference:
>
> [1] Liu et al.,  ARC: A Generalist Graph Anomaly Detector with In-Context Learning. NeurIPS 2024
>
> [2] Niu et al., Zero-shot Generalist Graph Anomaly Detection with Unified Neighborhood Prompts. IJCAI 2025
>
> ---
>
> > Q2. Is the loss from Equation 13 novel? What is the motivation?
>
> The proposed loss function in Equation 13 is motivated by the need for robust zero-shot cross-domain graph anomaly detection, where labels in the test graphs are unavailable and graphs from different domains exhibit heterogeneous structures and attributes.
>
> - The reconstruction loss ($\mathcal{L}_{\text{recon}}$) serves a dual purpose: it maximizes similarity between reconstructed and original embeddings for normal nodes, ensuring they are well-represented by the cross-domain dictionary, and minimizes similarity for anomalous nodes, amplifying the anomaly signal.
>
> - The triplet loss ($\mathcal{L}_{\text{triplet}}$) further strengthens the embeddings by explicitly pushing anomalous nodes away from normal nodes while considering both attribute-level and structure-level representations, ensuring that anomalies remain distinguishable even if partial reconstruction occurs.
>
> - Each component of the loss is necessary: the reconstruction term captures **instance-level** conformity to normal patterns, while the triplet term enforces **relational separation** between normal and anomalous nodes.
>
> The novelty lies in the joint integration of cross-domain dictionary-based reconstruction, truncated attention for filtering representative normal patterns, and triplet loss on both attribute and structure embeddings, enabling effective zero-shot anomaly detection across heterogeneous graphs without retraining or requiring test labels.
>
> ---
>
> > Q3. What are the values (numbers) in Figure 3?
>
> Below are the numerical results and we updated Figure 3 to include the values
>
> OWLEYE | OWLEYE-S | OWLEYE-N | OWLEYE-T |
> |-------------|-------------|-------------|-------------|
> | 35.86 | 35.06 | 34.98 | 34.08 |
>
> ---

---

> ### Author Response · Authors · 2025-11-25
> **Rebuttal Answer from Authors (Part II)**
>
> > Rest of W1: Figure 1 is hard to read because of the small font and markers, and the overwhelming amount of information overall. The related work is very brief and doesn’t cover relevant literature, such as cross-domain graph anomaly detection, which can be unsupervised.
>
>
> First, we have updated Figure 1 by
> - increasing the size of the font and markers
> - changing the color to better readiness
> - adding more legends to highlight the information
> - adding more context in the caption to illustrate
>
>
> Second, below are the more related works that have been added to the Section 4 and Appendix B. If you have additional relation work suggestions that we should discuss in the paper, please let us know.
> - _``Cross-domain graph anomaly detection (CD-GAD) has recently drawn growing interest as models trained on one graph often degrade when deployed on graphs with different structures or feature distributions. Early work by (Ding et al., 2021) introduced one of the first CD-GAD frameworks by aligning latent representations across source and target graphs to mitigate distributional shifts. More recently, Wang et al. (2023) proposed an anomaly-aware contrastive alignment approach that explicitly incorporates anomaly signals into cross-domain representation learning, improving robustness under heterogeneous domains. Complementary to these alignment-based methods, Pirhayatifard & Silva advanced a test-time adaptation framework that leverages homophily-guided self-supervision to adjust model parameters on the target graph without requiring labeled anomalies.''_
>
> ---
>
> > Q4 (W2). What information about the test graph is available in the problem setting considered?
>
> Here, we would like to clearly list and strengthen the problem setting as described in the paper.
>
> - First, all graphs in the test set are unavailable in the training phase.
>
> - In the training stage, only graphs in the training set are available and the graphs in the test set are not available. We store the patterns extracted from each training graph.
>
> - In the testing stage, the graphs in the training set are unavailable but the extracted patterns from these graining graphs are available.
>
> - The graphs in the test set are available for evaluation.
>
> - For example, in the 10-shot setting, 10 labeled examples are considered as the extracted patterns in the dictionary and we use these patterns for feature construction.
>
> ---
>
> > Q5 (W3). What is the replacement used for each component of OWLEYE in the ablation study? Are these components the state of the art?
>
> In the ablation study, we examine that replacing truncated attention with the attention method used in ARC [1] reduces the performance. OWLEYE is our full version, and OWLEYE-T is the ablation that replaces the truncated attention. ARC is the state-of-the-art method published in last year NeurIPS. OWLEYE-N and OWLEYE-S are two variants of our method by removing feature normalization (equation 4) and structural patterns (equation 8), respectively.
>
> Comparing OWLEYE with OWLEYE-N and OWLEYE-S aim to show the effectiveness of feature normalization (OWLEYE-N) and involvement of structural patterns (OWLEYE-S). Furthermore, in Table 11 in the appendix, we examine whether our feature alignment method could benefit ARC by replacing its feature preprocessing method with our method. The results show that our feature alignment method improves the performance of ARC by 1% w.r.t. AUPRC, indicating that our method is better than the feature preprocessing method used in ARC.
>
> Reference:
>
> [1] Liu et al.,  ARC: A Generalist Graph Anomaly Detector with In-Context Learning. NeurIPS 2024

---

> ### Author Response · Authors · 2025-11-25
> **Rebuttal Answer from Authors (Part III)**
>
> > Rest of W3. It is also unclear why 200 patterns are sufficient to identify the anomalies; visualizing the patterns and anomalies could be helpful to understand what is happening (not clear if that is what is done in Fig. 1).
>
> Thank you for your suggestion.
>
> As shown in Table 5, increasing the number of patterns from 10 to 200 leads to a 0.55\% performance gain; and beyond 200, the improvement becomes marginal (less than 0.2\%) and the performance becomes stable even if we keep increasing the dictionary size. These findings demonstrate that OWLEYE benefits from a larger pattern dictionary up to a saturation point, beyond which returns diminish.
>
> Moreover, we added corresponding visualizations in Appendix A.10 (Figure 7 and 8) to demonstrate that the interpretability of our method via the attention map by visualizing the correct prediction on normal node and anomalous node and the incorrect prediction on normal node and anomalous node on the Cora and Amazon datasets.
> - In the visualization of attention maps, lighter colors such as light green and light yellow consistently indicate high similarity to the patterns associated with normal nodes, as reflected in the label matrices in Figure 7 and 8.
> - By examining these visualizations, we observe a clear and consistent relationship between the attention intensity and the correctness of the model’s predictions:
>   - when the model correctly identifies a normal node, its attention map is dominated by light colors, suggesting a strong similarity to normal patterns stored in the dictionary;
>   - when it correctly identifies an anomalous node, the attention map becomes noticeably darker, indicating low similarity to normal behavior.
> - Importantly, this trend reverses for misclassified nodes: normal nodes that are wrongly predicted as anomalies exhibit darker color in attention map, while misclassified anomalous nodes show lighter colors, showing the high similarity to those of normal nodes.
> - This systematic behavior demonstrates that the attention map offers an intuitive and faithful interpretation mechanism, as the color patterns directly reflect whether the node under consideration resembles the learned normal patterns, thereby revealing both the reasoning behind correct predictions and the failure modes behind incorrect ones.
>
>
> ---
>
> > Rest of W3. There is no discussion on training time, which could be a deciding factor for practitioners.
>
> Following the experiment on efficiency analysis shown in Figure 3, ACM dataset is selected in the experiment for the better comparison between training time and fine-tuning time. In addition, though ACM is not the largest graph dataset in the experiment, it is the fair dataset to report all baselines’ efficiency performance because some baseline methods (like TAM and CARE) can run out of memory in larger datasets.
>
> Taking ACM as the example, we report the training time on the ACM dataset for all baseline methods in the table. The results show that the training time of our method is more efficient than most of the baseline methods with overall first place performance gain reported in Tables 1 and 2. We have added this experiment in our paper in Appendix A.11.
>
> | Method      | Training Time (s) |
> |-------------|-------------------|
> | BWGNN       | 16.6              |
> | GHRN        | 8.72              |
> | SLGAD       | 280               |
> | Dominant    | 470               |
> | TAM         | 254.71            |
> | CARE        | 550.23            |
> | ARC         | 1.89              |
> | UNPrompt    | 21.23             |
> | OWLEYE (ours)  | 3.84              |

---

### Official Review · Reviewer_wzbY · 2025-10-31

**Soundness:** 3
**Presentation:** 3
**Contribution:** 3
**Rating:** 6
**Confidence:** 4

**Summary:**

This paper proposes OWLEYE, a zero-shot cross-domain graph anomaly detection (GAD) model designed to establish a universal GAD framework. The core contribution lies in designing a novel cross-domain feature alignment module to address the limitations of existing generalist models when aligning graph data across different domains. Additionally, the model identifies anomalies through dictionary learning and a truncated attention-based reconstruction mechanism. Experimental results demonstrate superiority over baseline methods across multiple datasets.

**Strengths:**

1. Zero-shot cross-domain graph anomaly detection is a significant and practical problem. The paper decomposes this challenge into three stages—feature alignment, pattern learning, and anomaly detection, addressing each sequentially to ensure a technically sound overall solution.
2. The paper clearly identifies the shortcomings of existing general GAD models in feature alignment. It proposes a well-motivated, novel, and effective solution.
3. Extensive experiments across multiple datasets demonstrate that OWLEYE significantly outperforms various state-of-the-art baselines. The case study on continuous learning also preliminarily shows the model's potential to absorb new knowledge in a plug-and-play manner.

**Weaknesses:**

1. The overall innovation of the framework is incremental. Its technical pipeline from feature alignment and multi-hop residual aggregation to a multi-pattern dictionary similar to “context learning” largely follows the established paradigm of generalist GAD models.
2. Section 2.2 thoroughly argues for using “only structural similarity” (Equation 10) to address “camouflaged” anomalies. However, the final reconstruction formula (12) employs an undefined attribute-based similarity `sim(G, Dict_H)` during attribute reconstruction, contradicting prior descriptions (its own design rationale).
3. The paper fails to explicitly describe the specific implementation of the “standard attention” variant in ablation study, nor does it provide a concrete analysis of its significant performance decline.

**Questions:**

1. The paper employs PCA for feature projection but offers no justification for choosing this method over other nonlinear or domain adaptation techniques. Furthermore, how is the situation handled if the feature dimension of the original graph is smaller than the preset projection dimension `d`?

2. During inference, the dictionary size and the `k` value in truncated attention are critical hyperparameters. Could the paper provide guiding principles or sensitivity analysis for selecting these parameters for new tasks? Were inference hyperparameters unified across all test sets in comparative experiments?

---

> ### Author Response · Authors · 2025-11-25
> **Rebuttal Answer from Authors (Part I)**
>
> Dear Reviewer wzbY,
>
> Thanks very much for your review and appreciation of our paper’s significant contribution in systematical design, shortcoming analysis of existing work, and extensive experiments.
>
> Your suggestions are very actionable, we take each seriously and prepare the follow-up answer. The corresponding parts are updated in the pdf file and marked with the orange background color.
>
> > W1. The overall innovation of the framework is incremental. Its technical pipeline from feature alignment and multi-hop residual aggregation to a multi-pattern dictionary similar to “context learning” largely follows the established paradigm of generalist GAD models.
>
> We would like to emphasize that the **key novelty lies in how to enable robust zero-shot and continual learning** that most recent generalist GAD methods (e.g., ARC [1] and UNPrompt [2]) fail to consider one aspect or both. For example,
> - ARC [1] can not support zero-shot or continual learning as it could not adapt to a new domain without retraining on all previous ones, but our method builds a reusable pattern dictionary and an alignment module, allowing direct deployment on unseen graphs and incremental expansion without retraining.
> - UNPrompt [2] can not perform continual learning because it heavily relies on prompt tuning, which must be recalibrated whenever a new domain is observed, but our method maintains a structured, persistent dictionary of attribute-level and structure-level patterns, enabling stable zero-shot inference and continual knowledge accumulation.
>
> To be more specific, our method does not assume any label availability in the test graph, making it applicable in realistic scenarios where anomaly labeling is costly or infeasible.
> - Cross-domain feature alignment using pairwise distance statistics preserves structural patterns and semantic integrity across heterogeneous graphs.
> - Multi-domain pattern dictionaries allow the model to continually integrate new knowledge from additional graphs and incrementally update normal and abnormal patterns without retraining, a capability absent in prior work.
> - Truncated attention-based reconstruction filters out potential abnormal nodes during inference, enabling accurate anomaly detection in unseen graphs in zero-shot setting.
>
> Therefore, our framework goes beyond incremental technical steps and introduces mechanisms for zero-shot detection and continual learning across heterogeneous graph domains, which fundamentally extends the generalist GAD paradigm.
>
> Reference:
>
> [1] Liu et al.,  ARC: A Generalist Graph Anomaly Detector with In-Context Learning. NeurIPS 2024
>
> [2] Niu et al., Zero-shot Generalist Graph Anomaly Detection with Unified Neighborhood Prompts. IJCAI 2025
>
> ---

---

> > ### Author Response · Authors · 2025-11-25
> > **Rebuttal Answer from Authors (Part II)**
> >
> > > W2. Section 2.2 thoroughly argues for using “only structural similarity” (Equation 10) to address “camouflaged” anomalies. However, the final reconstruction formula (12) employs an undefined attribute-based similarity sim(G, Dict_H) during attribute reconstruction, contradicting prior descriptions (its own design rationale).
> >
> > In brief, the two similarities serve distinct purposes rather than contradict each other. In other words,
> > - yes, we use structural similarity (Equation 10) only to **govern which domain patterns to trust**,
> > - whereas the latter attribute similarity (Equation 12) **refines how attributes are reconstructed**.
> >
> > Also, we updated the definition of attribute-based similarity sim(G, Dict_H) in lines 250-252.
> >
> > To extend the above statement, we would like to clarify more that
> > - (1) including both attribute-level representation and structural-level representation indeed help successfully identify more anomalies, which is validated in the ablation study in Figure 3 (OWLEYE vs OWLEYE-S),
> > - and (2) in the new experiment below (also added in Appendix A.12), we figure out that using both structural and attribute similarity for domain similarity measurement is less stable than relying on structural similarity alone, because camouflaged anomalies may mimic normal neighbors’ attributes and cross-domain feature discrepancies make reliable measurement more challenging.
> >
> > Below are the new experimental results comparing using both structural similarity and attribute similarity (A+S) for domain similarity measurement vs only using structural similarity (S-Only) for domain similarity measurement.
> >
> > | Dataset      | A+S (%)                  | S-Only (%)               | Improvement (%) |
> > |--------------|---------------------------|---------------------------|------------------|
> > | Cora         | 43.26 ± 0.54          | 43.94 ± 0.46          | 0.68           |
> > | Flickr         | 37.83 ± 0.39          | 37.69 ± 0.25          | -0.14          |
> > | ACM          | 39.84 ± 0.28          | 39.75 ± 0.13          | -0.09          |
> > | BlogCatalog  | 34.34 ± 0.53          | 34.99 ± 0.31          | 0.65           |
> > | Facebook     | 6.11 ± 1.35           | 5.62 ± 1.17           | -0.49          |
> > | Weibo          | 58.61 ± 5.18          | 60.90 ± 0.21          | 2.28           |
> > | Reddit          | 4.05 ± 0.12           | 4.25 ± 0.11           | 0.20           |
> > | Amazon       | 48.01 ± 18.44         | 62.20 ± 3.18          | 14.19          |
> > | Average      | 34.26 ± 3.35          | 36.17 ± 0.73          | 2.16           |
> >
> >
> > | Dataset      | No Structural Patterns (AUPRC %) | Our Method (AUPRC %) | Improvement (%) |
> > |--------------|----------------------------------|-----------------------|------------------|
> > | Cora         | 39.38 ± 1.32                     | 43.94 ± 0.46          | 4.56             |
> > | Flickr       | 38.06 ± 0.08                     | 37.69 ± 0.25          | -0.36            |
> > | ACM          | 39.64 ± 0.15                     | 39.75 ± 0.13          | 0.11             |
> > | BlogCatalog  | 35.42 ± 0.26                     | 34.99 ± 0.31          | -0.43            |
> > | Facebook     | 4.71 ± 0.27                      | 5.62 ± 1.17           | 0.90             |
> > | weibo        | 57.18 ± 1.09                     | 60.90 ± 0.21          | 3.72             |
> > | Reddit       | 4.12 ± 0.15                      | 4.25 ± 0.11           | 0.13             |
> > | Amazon       | 61.98 ± 1.08                     | 62.20 ± 3.18          | 0.22             |
> > | Average      | 35.06 ± 0.55                     | 36.17 ± 0.73          | 1.11             |
> >
> >
> > The experimental results in the first table show that using both structural similarity and attribute similarity (A+S) for domain similarity measurement decreases the performance. The second table shows that including structural patterns indeed increases the performance.
> >
> > ---
> >
> > > W3. The paper fails to explicitly describe the specific implementation of the “standard attention” variant in ablation study, nor does it provide a concrete analysis of its significant performance decline.
> >
> > In the ablation study, “standard attention” refers to the self-attention mechanism in Transformer [1], which is widely used in in-context learning.
> >
> > In the ablation study, we compare truncated attention with standard attention mechanism (OWLEYE-T). The results show that when we replace our proposed truncated attention with the attention mechanism used in in-context learning, the overall performance drops by more than around 1.8\% across eight test graphs.
> >
> > Reference:
> >
> > [1] Ashish, et al.,  Attention is all you need. NeurIPS 2017
> >
> > ---

---

> ### Author Response · Authors · 2025-11-25
> **Rebuttal Answer from Authors (Part III)**
>
> > Q1. The paper employs PCA for feature projection but offers no justification for choosing this method over other nonlinear or domain adaptation techniques. Furthermore, how is the situation handled if the feature dimension of the original graph is smaller than the preset projection dimension d?
>
> We added more linear and nonlinear feature projection methods including PCA, SVD, Kernel PCA, and NMF. The experimental results show that our method with PCA still achieves the best performance as modeled in the paper.
>
> Comparing PCA with nonlinear methods like Kernel PCA and NMF, we observe that using more complicated feature projection does not necessarily improve the performance, as it might distort and misalign the original feature space, leading to performance drop. (We have added the following results in Appendix A.13.)
>
> | Dataset       | PCA (AUPRC %)       | SVD (AUPRC %)       | Kernel PCA (AUPRC %) | NMF (AUPRC %)|
> |---------------|------------------|------------------|--------------------|--------------------|
> | Cora          | 43.94 ± 0.46         | 44.13 ± 0.81         | 44.05 ± 0.68           | 15.06 ± 2.28           |
> | Flickr        | 37.69 ± 0.25         | 38.18 ± 0.37         | 37.54 ± 0.46           | 33.09 ± 0.70           |
> | ACM           | 39.75 ± 0.13         | 39.18 ± 0.22         | 38.75 ± 0.15           | 32.28 ± 1.13           |
> | BlogCatalog   | 34.99 ± 0.31         | 35.28 ± 0.31         | 34.93 ± 0.24           | 33.36 ± 0.48           |
> | Facebook      | 5.62 ± 1.17          | 5.00 ± 0.27          | 5.63 ± 1.05            | 7.43 ± 1.34            |
> | Weibo         | 60.90 ± 0.21         | 57.89 ± 2.48         | 60.55 ± 0.22           | 49.90 ± 3.66           |
> | Reddit        | 4.25 ± 0.11          | 3.44 ± 0.24          | 4.10 ± 0.15            | 3.42 ± 0.07            |
> | Amazon        | 62.20 ± 3.18     | 44.27 ± 3.45     | 38.75 ± 3.07       | 20.61 ± 7.57           |
> | Average   | **36.17 ± 0.73**     | 33.42 ± 1.02     | 33.04 ± 0.75       | 24.39 ± 2.15           |
>
>
> When the feature dimension is smaller than the preset projection dimension, we use Gaussian Random Projection [1] to 256 following ARC [2] and then do the feature reduction.
>
> Reference:
>
> [1] Dimitris Achlioptas: Database-friendly random projections: Johnson-Lindenstrauss with binary coins. J. Comput. Syst. Sci. 2003
>
> [2] Liu et al.,  ARC: A Generalist Graph Anomaly Detector with In-Context Learning. NeurIPS 2024

---

> ### Author Response · Authors · 2025-11-25
> **Rebuttal Answer from Authors (Part IV)**
>
> > Q2. During inference, the dictionary size and the k value in truncated attention are critical hyperparameters. Could the paper provide guiding principles or sensitivity analysis for selecting these parameters for new tasks? Were inference hyperparameters unified across all test sets in comparative experiments?
>
> In brief, the inference hyperparameters are unified across all test sets. We include the hyperparameter analysis in Table 10 and the results show that our method is stable to most of the hyperparameters.
>
> For the dictionary size and k value you mentioned, we did further explanation and additional analysis below. This new experiment and corresponding analysis are added in Appendix A.14.
>
> As for the dictionary size, we analyze how varying the dictionary size impacts the performance in Table 5. The experimental result shows that a larger value of dictionary size leads to better performance. Thus, we can select a large value of dictionary size for better performance.
>
> In the experiment, $k$ is a relative value according to the dictionary size.
> - We set the value of $k$ to $p$% of patterns in the dictionary, where $n_{sup}$=1000 for each graph.
> - We vary the percentage of it from 5% to 50% as well as two specific numbers of k (k=5 and k=10) and report the overall results across eight graphs below.
>
> To be more specific, in the below new experiment, the range of the value of $k$ is based on the observation that the percentage of the anomaly is usually lower than 10%. Using the ratio of dictionary size can filter out most of the anomalies and then reduce the uncertainty.
> - In a situation where the percentage of anomalies is unusually high, we can manually increase the value of $k$. The results show that when we set the value of $k$ to 10% to 30% of patterns in the dictionary, the AUPRC scores are around 36.1% and its performance decreases if we increase the percentage to 50%.
> - One possible explanation is that we filter out too many patterns that might be useful for identifying anomalies. Similarly, when we decreased the value of k to a value that could not filter out most of the anomalies, the performance starts to decrease, as the method involves the anomalies for feature construction.
>
>
> | Dataset     | k=5               | k=10              | k=0.05*n_{sup}        | k=0.1*n_{sup}          | k=0.2*n_{sup}          | k=0.3*n_{sup}          | k=0.5*n_{sup}         |
> |-------------|-------------------|-------------------|-----------------------|-----------------------|-----------------------|-----------------------|-----------------------|
> | Cora        | 0.4389 ± 0.0153   | 0.4418 ± 0.0082   | 0.4394 ± 0.0067       | 0.4408 ± 0.0073       | 0.4404 ± 0.0105       | 0.4355 ± 0.0090       | 0.4393 ± 0.0047       |
> | Flickr      | 0.3752 ± 0.0057   | 0.3774 ± 0.0038   | 0.3762 ± 0.0029       | 0.3766 ± 0.0030       | 0.3762 ± 0.0034       | 0.3823 ± 0.0026       | 0.3769 ± 0.0026       |
> | ACM         | 0.3973 ± 0.0011   | 0.3973 ± 0.0009   | 0.3974 ± 0.0014       | 0.3976 ± 0.0010       | 0.3973 ± 0.0014       | 0.3914 ± 0.0011       | 0.3974 ± 0.0012       |
> | BlogCatalog | 0.3475 ± 0.0033   | 0.3497 ± 0.0034   | 0.3500 ± 0.0032       | 0.3493 ± 0.0028       | 0.3491 ± 0.0059       | 0.3505 ± 0.0037       | 0.3499 ± 0.0031       |
> | Facebook    | 0.0573 ± 0.0115   | 0.0567 ± 0.0117   | 0.0561 ± 0.0116       | 0.0553 ± 0.0122       | 0.0516 ± 0.0120       | 0.0587 ± 0.0119       | 0.0561 ± 0.0117       |
> | Weibo       | 0.5973 ± 0.0256   | 0.5931 ± 0.0280   | 0.5952 ± 0.0246       | 0.6063 ± 0.0249       | 0.6097 ± 0.0085       | 0.6028 ± 0.0068       | 0.6094 ± 0.0021       |
> | Reddit      | 0.0407 ± 0.0018   | 0.0405 ± 0.0007   | 0.0406 ± 0.0008       | 0.0406 ± 0.0009       | 0.0406 ± 0.0010       | 0.0392 ± 0.0008       | 0.0405 ± 0.0009       |
> | Amazon      | 0.6109 ± 0.0353   | 0.6126 ± 0.0347   | 0.6196 ± 0.0323       | 0.6204 ± 0.0234       | 0.6220 ± 0.0312       | 0.6326 ± 0.0248       | 0.6191 ± 0.0305       |
> | **Average** | **0.3581 ± 0.0125** | **0.3586 ± 0.0114** | **0.3593 ± 0.0104**   | **0.3609 ± 0.0094**   | **0.3609 ± 0.0093**   | **0.3616 ± 0.0076**   | **0.3611 ± 0.0071**   |

---

### Meta-Review · Area_Chair_SAse · 2025-12-30

**Summary:**

The paper proposes OWLEYE, a zero-shot framework for cross-domain graph anomaly detection based on feature alignment, multi-domain pattern dictionaries, and truncated-attention reconstruction.

The submission received four reviews. Two confident reviewers (confidence 4) recommended acceptance with a score of 6. A third reviewer initially scored the paper as 4 but explicitly stated after the discussion that they would raise the score to 6. The fourth reviewer provided a critical review with a score of 2.

Some concerns raised by the critical reviewer, in particular the reliance on heuristic design choices without strong theoretical guarantees (W1), are inherent to the approach and difficult to fully address. I largely agree with this limitation. Other concerns, however, were addressed through the rebuttal, including feature alignment robustness and normalization (W2, W6), clarity and notation (W5), and to some extent sensitivity and stability aspects (W3). The critique regarding insufficient baseline comparisons (W4) remains vague, as no concrete missing baselines were identified.

Overall, three out of four reviewers support acceptance based on the methodological design and extensive empirical evaluation. Despite limited theoretical grounding, the paper makes a solid and practically relevant contribution to cross-domain graph anomaly detection and should be accepted.

**Reviewer Concerns:**

Limited theoretical grounding; several design choices are heuristic

Assumptions on cross-domain pattern transferability may not always hold

Baseline coverage questioned, though critique is vague and lacks concrete alternatives

**Reviewer Scores:**

wzbY: stays at 6

fw57: stays at 6

aCfo: increase from 4 to 6 (explicitly stated)

fjeF: likely stays at 2 (or at most 4), 3.5 out 6 concerns addresses, 2.5 remain.

---

### Decision · Program_Chairs · 2026-01-26

Accept (Poster)